The EMBO Journal (2013) 32, 3161–3175
www.embojournal.org

# Isogenic human pluripotent stem cell pairs reveal the role of a KCNH2 mutation in long-QT syndrome

**Milena Bellin[1,*], Simona Casini[1,7], Richard P Davis[1,2,7], Cristina D'Aniello[1,7], Jessica Haas[3], Dorien Ward-van Oostwaard[1], Leon GJ Tertoolen[1], Christian B Jung[3], David A Elliott[4], Andrea Welling[5,6], Karl-Ludwig Laugwitz[3,6], Alessandra Moretti[3,6,*] and Christine L Mummery[1]**

[1]Department of Anatomy and Embryology, Leiden University Medical Center, Leiden, The Netherlands, [2]Netherlands Proteomics Institute, Utrecht, The Netherlands, [3]I. Medizinische Klinik und Poliklinik, Klinikum rechts der Isar der Technischen Universität München, Munich, Germany, [4]Murdoch Childrens Research Institute, Royal Childrens Hospital, Parkville, Victoria, Australia, [5]Institut für Pharmakologie und Toxikologie der Technischen Universität München, Munich, Germany and [6]DZHK (German Centre for Cardiovascular Research)—Partner Site Munich Heart Alliance, Munich, Germany

**Patient-specific induced pluripotent stem cells (iPSCs) will assist research on genetic cardiac maladies if the disease phenotype is recapitulated _in vitro_. However, genetic background variations may confound disease traits, especially for disorders with incomplete penetrance, such as long-QT syndromes (LQTS). To study the LQT2-associated c.A2987T (N996I) _KCNH2_ mutation under genetically defined conditions, we derived iPSCs from a patient carrying this mutation and corrected it. Furthermore, we introduced the same point mutation in human embryonic stem cells (hESCs), generating two genetically distinct isogenic pairs of LQTS and control lines. Correction of the mutation normalized the current ($I_{Kr}$) conducted by the HERG channel and the action potential (AP) duration in iPSC-derived cardiomyocytes (CMs). Introduction of the same mutation reduced $I_{Kr}$ and prolonged the AP duration in hESC-derived CMs. Further characterization of N996I-HERG pathogenesis revealed a trafficking defect. Our results demonstrated that the c.A2987T _KCNH2_ mutation is the primary cause of the LQTS phenotype. Precise genetic modification of pluripotent stem cells provided a physiologically and functionally relevant human cellular context to reveal the pathogenic mechanism underlying this specific disease phenotype.**

*Corresponding authors. M Bellin, Department of Anatomy and Embryology, Leiden University Medical Center, Einthovenweg 20, 2333 ZC Leiden, The Netherlands. Tel.: +31 715269382; Fax: +31 715268289; E-mail: m.bellin@lumc.nl or A Moretti, I. Medical Department, Klinikum rechts der Isar, Technical University of Munich, Cardiology, Ismaninger Strasse 22, 81675 Munich, Germany. Tel.: +49 8941406907; Fax: +49 8941404901; E-mail: amoretti@med1.med.tum.de
[7]These authors contributed equally to this work.

The EMBO Journal (2013) **32,** 3161–3175. doi:10.1038/emboj.2013.240; Published online 8 November 2013
Subject Categories: development; molecular biology of disease
Keywords: gene targeting; HERG; human embryonic stem cells; induced pluripotent stem cells; long-QT syndrome

## Introduction

Transgenic animal models of human cardiac diseases and cultured cells have been crucial to cardiovascular research, contributing to both our basic understanding of normal development and disease mechanisms. However, these approaches often only partially recapitulate the molecular and cellular phenotypes observed in patients because human heart physiology, as well as the cardiomycyte's gene and protein expression profile, differs from almost all other experimentally accessible species (Davis _et al_, 2011).

Human induced pluripotent stem cells (hiPSCs) (Takahashi _et al_, 2007b; Yu _et al_, 2007) are creating exciting new opportunities for biomedical research by providing platforms to study mechanisms of genetic disease pathogenesis that could lead to new therapies or reveal drug sensitivities (Braam _et al_, 2013; Liang _et al_, 2013). Patient-specific hiPSCs that carry all disease-relevant genetic alterations are important not only for understanding monogenic and complex disease mechanisms in the affected cell types _in vitro_ and identifying potential treatments (Gold-von Simson _et al_, 2009; Lee _et al_, 2009; Marchetto _et al_, 2010; Axelrod _et al_, 2011; Pini _et al_, 2012), but also for providing insights into factors that predispose individuals to develop overt symptoms. However, the application of hiPSCs in disease modelling has been limited by the paucity of experimental tools for distinguishing subtle but disease-relevant phenotypic changes from background-related variations.

Inherited long-QT syndrome (LQTS) is a life-threatening, often autosomal dominant, disorder characterized by prolonged ventricular repolarization, a propensity to polymorphic ventricular tachycardia and sudden cardiac death in young patients (Crotti _et al_, 2008). Among genotyped individuals, $\sim 90\%$ of affected patients have mutations in the genes encoding repolarizing $K^+$ channels of the delayed rectifier currents, $I_{Ks}$ (_KCNQ1_ in LQT1) and $I_{Kr}$ (_KCNH2_ in LQT2) (Morita _et al_, 2008). LQTS presents clinically with a broad range of phenotypes even among family members with identical mutations (Giudicessi and Ackerman, 2013). This heterogeneity is thought to result from the interplay of complex networks of direct and indirect causal factors, including epistatic effects of genetic background, resulting in incomplete penetrance and variable severity. LQTS was one of the first cardiac diseases recapitulated using hiPSCs as

a model (Moretti *et al*, 2010; Itzhaki *et al*, 2011; Matsa *et al*, 2011; Yazawa *et al*, 2011; Davis *et al*, 2012; Egashira *et al*, 2012; Lahti *et al*, 2012). However, there is a growing realization that comparing patient cells to unrelated healthy control cells, no matter how they are matched, does not control for any effects of genetic modifier loci (Bellin *et al*, 2012; Cherry and Daley, 2013; Sinnecker *et al*, 2013). Furthermore, additional key factors may contribute to variability among individual hiPSC clones, including the number and location of viral integrations following reprogramming (Gonzalez *et al*, 2011), genetic heterogeneity in the starting somatic cell population (Gore *et al*, 2011), copy number variations (Panopoulos *et al*, 2011; Abyzov *et al*, 2012), and their epigenetic status (Nazor *et al*, 2012). For these reasons, it is preferable to compare cell lines in which the disease-causing genetic lesion of interest is the only modified variable. Gene correction in hiPSCs from patients with known genetic mutations is a powerful tool for overcoming the limitations imposed by individual variability among independent controls, and with recent technological improvements genome editing is now feasible in both human embryonic stem cells (hESCs) and hiPSCs (Costa *et al*, 2007; Zou *et al*, 2009; Hockemeyer *et al*, 2011; Asuri *et al*, 2012). The generation of genetically matched controls that differ exclusively in well-validated susceptibility variants for maladies is a generally applicable solution to the key problem of distinguishing pathologically relevant phenotypic changes from other background-related variations (Soldner *et al*, 2011; Reinhardt *et al*, 2013). However, to date, no isogenic cardiac disease iPSC lines have been reported, and neither has the disease phenotype been investigated in parallel in both hiPSCs and hESCs harbouring the same mutation.

In this study, we sought to generate and characterize a panel of control and LQT2-related human pluripotent stem cell (hPSC) lines. In order to investigate the specific effects of the c.A2987T (N996I) *KCNH2* mutation without the confounding elements that could result from individual genetic background variability, we studied this mutation under two genetically defined conditions. First, by genetically correcting the *KCNH2* mutation in the LQT2-hiPSCs derived from a patient, and second, by introducing the same c.A2987T (N996I) mutation into an *NKX2.5*$^{eGFP/w}$ hESC reporter line that allows the cardiac cells to be selected from a mixed differentiated population (Elliott *et al*, 2011). Cardiomyocytes (CMs) differentiated from these two pairs of isogenic hPSCs demonstrated that the N996I *KCNH2* mutation causes an ∼30–40% $I_{Kr}$ reduction with consequential action potential (AP) duration (APD) prolongation, and indicated a trafficking defect as the associated pathological mechanism responsible for the disease electrophysiological phenotype.

## Results

### Generation of hiPSCs with c.A2987T (N996I) KCNH2 mutation

Dermal fibroblasts were obtained from a 38-year-old woman with a diagnosis of familial type-2 LQTS (Supplementary Figure S1A and B). Genetic screening indicated a heterozygous c.A2987T mutation in exon 13 of the *KCNH2* gene (Figure 1A), resulting in an asparagine to isoleucine substitu-

tion at position 996 (N996I) within the intracellular C-terminal region of the encoded HERG potassium channel (Figure 1B). We generated hiPSCs (LQT2-hiPSCs$^{N996I}$) from the primary skin fibroblasts by retroviral transduction of four reprogramming genes (*OCT4*, *SOX2*, *KLF4*, and *MYC*) (Takahashi *et al*, 2007a) (Figure 1C), and three of the resulting clones underwent further characterization. These clones expressed the hESC markers SSEA4 and NANOG, as detected by immunofluorescence staining (Figure 1D). Additionally, qRT–PCR indicated that the hiPSC lines had reactivated the endogenous pluripotency genes *OCT4*, *SOX2*, *LEFTYA*, *NANOG*, *REX1*, and *TDGF1*, and silenced the retroviral transgenes (Supplementary Figure S2A and B). Finally, differentiation of the hiPSCs as embryoid bodies demonstrated the capacity of all lines to generate derivatives of the three embryonic germ layers *in vitro* (Supplementary Figure S3). On the basis of these findings, one hiPSC clone was selected for targeted genetic correction.

### Targeted gene correction in LQT2-hiPSCs$^{N996I}$ and targeted gene mutation in NKX2.5$^{eGFP/w}$ hESCs

The study design is illustrated in Figure 2A. A conventional homologous recombination strategy was performed for either targeted gene correction in LQT2-hiPSCs$^{N996I}$ or targeted gene mutation in the *NKX2.5*$^{eGFP/w}$ hESCs. We constructed two targeting vectors that included either a wild-type A or a mutated T nucleotide at *KCNH2* c.2987 (*KCNH2-A-loxP-pGK-Neo-loxP* and *KCNH2-T-loxP-pGK-Neo-loxP*, respectively) (Figure 2B). Targeting of either the mutant allele (hiPSCs) or one of the two wild-type alleles (hESCs) is expected to result in a single base pair change that either repairs (LQT2-hiPSCs$^{corr}$) or introduces (*NKX2.5*$^{eGFP/w}$ hESCs$^{N996I}$) the N996I HERG mutation. The coding regions and the exon–intron boundaries of *KCNH2* in the hiPSCs and hESCs were sequenced to exclude the presence of any other variants.

The linearized targeting constructs were electroporated into the hiPSCs and hESCs and G418-resistant colonies isolated. Targeted clones were identified using a PCR-based screen that amplified the novel junction fragment between the genomic DNA and the integrated targeting vector generated following homologous recombination (Figure 2B and C). These PCR products were sequenced to confirm homologous recombination and to determine whether the mutation was corrected or introduced. We obtained one corrected LQT2-hiPSC clone (LQT2-hiPSCs$^{corr}$) and one mutated *NKX2.5*$^{eGFP/w}$ hESC clone (LQT2-hESCs$^{N996I}$) (Figure 2D). The G418-resistance cassette was excised and the cell line cloned. Similar to the parental cell lines, the LQT2-hiPSCs$^{corr}$ and *NKX2.5*$^{eGFP/w}$ hESCs$^{N996I}$ expressed markers of undifferentiated hPSCs (Supplementary Figure S4A and B), could differentiate into the cellular derivatives of the three embryonic germ layers *in vitro* (Supplementary Figure S5), passed the PluriTest with a high 'pluripotency score' and a low 'novelty score', indicating that they resemble normal hPSCs (Supplementary Figure S6A), and retained a normal karyotype (Supplementary Figure S6B and C).

### The c.A2987T (N996I) KCNH2 mutation causes a long-QT phenotype in hPSC-derived CMs

The LQT2-hiPSCs$^{corr}$ and *NKX2.5*$^{eGFP/w}$ hESCs$^{N996I}$, together with their parental cell lines, were differentiated to CMs, with spontaneously contracting foci first observed at 10–12 days of

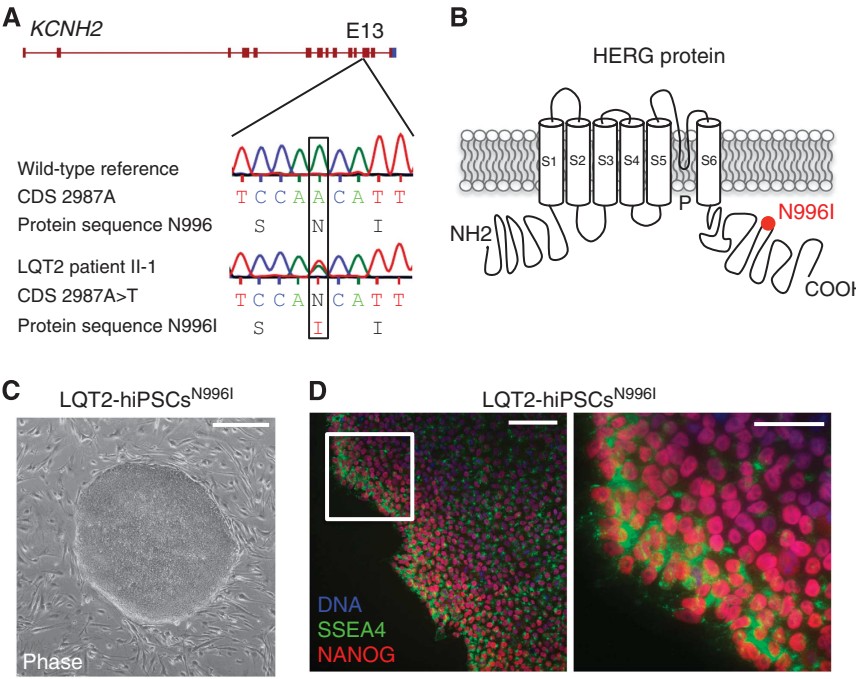

**Figure 1** Generation of hiPSCs from a patient with type-2 long-QT syndrome. (**A**) Genetic screening in the patient revealed the heterozygous single-nucleotide mutation A→T in exon 13 of the *KCNH2* gene, in position 2987 of the coding sequence (CDS) (c.A2987T, NM_000238.3), resulting in the substitution of an asparagine with an isoleucine at position 996 of the protein (N996I, NP_000229.1). (**B**) The N996I mutation (red dot) is located in the C-terminal of the HERG protein, which is made of six trans-membrane domains (S1–S6), an amino (NH2) domain, a carboxyl (COOH) domain, and a pore (P) region. (**C**) Example of a hiPSC colony harbouring the c.A2987T (N996I) *KCNH2* mutation (LQT2-hiPSCs^N996I). Scale bar: 400 μm. (**D**) Immunofluorescence analysis of pluripotency markers SSEA4 (green) and NANOG (red) in a representative LQT2-hiPSC^N996I clone, with nuclear staining (DNA, blue). The image on the right is a magnification of the area framed in the left image. Scale bars: 100 μm (left image); 50 μm (right image).

differentiation. These were mechanically micro-dissected and allowed to mature further (Moretti *et al*, 2010). Dissociated cells displayed characteristic sarcomeric structures that were positive for troponin I (TNNI) and alpha-actinin (Figure 3A), and continued to spontaneously contract. Quantitative RT–PCR in cardiac cells, corresponding to either micro-dissected beating areas (hiPSCs) or *NKX2.5* eGFP$^+$ cell populations (hESCs), revealed upregulation of both the *KCNH2-1a* and *-1b* transcript variants, as well as other ion channels essential for AP generation, compared to their undifferentiated counterparts (Figure 3B).

To confirm that the *loxP* sequence that remained in the *KCNH2* locus following excision of the selection cassette did not interfere with HERG expression, we took advantage of another LQT2-hiPSC clone that was targeted and contained the residual *loxP* sequence, but was not corrected (LQT2-hiPSCs^N996I/loxP-control). Analysing the response of the CMs to the selective $I_{Kr}$ channel blocker E-4031 using multielectrode arrays revealed that, consistent with earlier reports (Matsa *et al*, 2011), increasing concentrations of E-4031 caused prolongation of field potential duration (FPD) in all analysed LQT2-hiPSC lines (original LQT2-hiPSCs^N996I, LQT2-hiPSCs^N996I/loxP-control, and LQT2-hiPSCs^corr) (Supplementary Figure S7A). Importantly, the field potential response did not differ between LQT2-hiPSC^N996I and LQT2-hiPSC^N996I/loxP-control CMs and was prolonged when compared to LQT2-hiPSC^corr CMs, indicating not only that the N996I mutation conferred increased sensitivity to $I_{Kr}$ blockade but also that the remaining *loxP* sequence did not appear to interfere with their electrophysiological phenotype. We therefore used the LQT2-

hiPSCs^N996I/loxP-control line for further electrophysiological characterization. Moreover, similar analysis in CMs from *NKX2.5*^eGFP/w hESCs and *NKX2.5*^eGFP/w hESCs^N996I showed that introduction of the N996I mutation in a genetically distinct background resulted in a significant, E-4031-dependent FPD prolongation that was comparable to that measured in mutated LQT2-hiPSC CMs, suggesting an exclusive effect of this mutation on the observed phenotype (Supplementary Figure S7A–D).

To further investigate whether the correction of the c.A2987T (N996I) mutation resulted in an increase in the repolarizing potassium current specifically transduced by the HERG channel, we recorded $I_{Kr}$ in individual hiPSC-derived CMs. Typical examples of current traces recorded before and after addition of 1 μM of E-4031, in mutated and corrected LQT2-hiPSC-derived CMs are shown in Figure 4A and B, where $I_{Kr}$ is defined as E-4031-sensitive current. Average $I_{Kr}$ density in LQT2-hiPSC^N996I CMs was significantly smaller than in the LQT2-hiPSC^corr CMs (Figure 4C and D). At a membrane potential of −20 mV, both $I_{Kr}$ density (measured at the end of the test pulse) and peak-tail $I_{Kr}$ density were decreased by 33%. To confirm whether the c.A2987T (N996I) mutation is the cause of the LQT2 phenotype observed in hPSC-derived myocytes, we measured $I_{Kr}$ in CMs derived from *NKX2.5*^eGFP/w hESCs and *NKX2.5*^eGFP/w hESCs^N996I that represent a different genetic background. Representative examples of current traces recorded in wild-type and mutated hESC-CMs are shown in Figure 5A and B. Average $I_{Kr}$ density was significantly reduced in the mutated hESC-CMs compared with the wild-type cells (Figure 5C and D). At a membrane potential of −20 mV, $I_{Kr}$ density (measured at

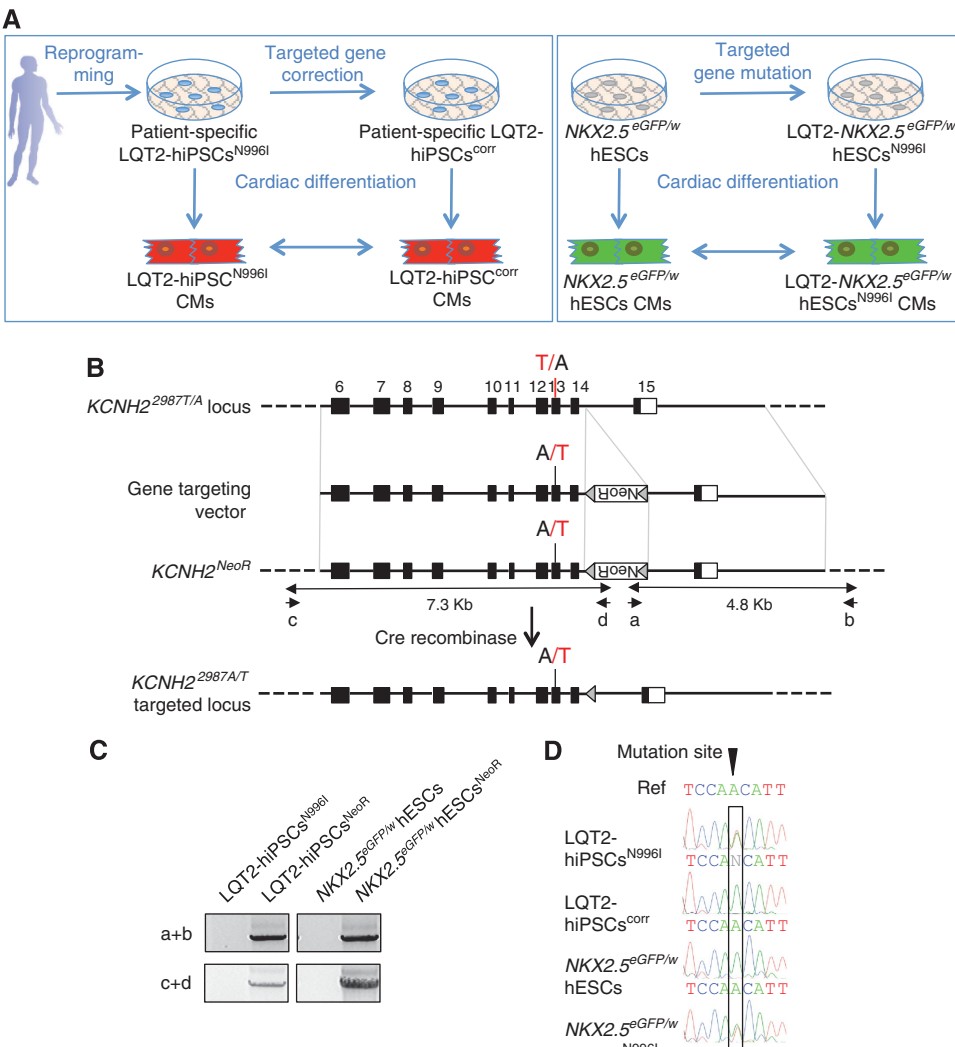

**Figure 2** Gene targeting by homologous recombination in LQT2-hiPSCs[N996I] and in *NKX2.5*[eGFP/w] hESCs. (**A**) Schematic showing the project rationale. Patient-specific LQT2-hiPSCs harbouring the N996I mutation were corrected and *NKX2.5*[eGFP/w] hESCs were mutated by gene targeting. Parental and genetically modified hPSC lines were differentiated into CMs and their electrophysiological phenotypes were analysed and compared. (**B**) The strategy for precise genomic modification of *KCNH2*. Top line, structure of the *KCNH2* locus. Numbered black boxes indicate exons 6–15. Exon 13 is mutated (red T) in LQT2-hiPSCs[N996I] and wild type (black A) in *NKX2.5*[eGFP/w] hESCs. The gene targeting vector for correcting the mutation in LQT2-hiPSCs has the wild-type adenine nucleotide, whereas the gene targeting vector for introducing the mutation in *NKX2.5*[eGFP/w] hESCs has the mutated thymine nucleotide. NeoR, the PGK-Neo cassette encoding G418 resistance flanked by *loxP* sequences (grey triangles), was inserted in the reverse direction. PCR primers (a, b) and (c, d) were used to identify the targeted clone. (**C**) PCR analysis using these primers generated specific bands of 4.8 kb (5′ homology arm) and 7.3 kb (3′ homology arm) from targeted clones (LQT2-hiPSCs[NeoR] and *NKX2.5*[eGFP/w] hESCs[NeoR]). (**D**) Sequence analysis of PCR-amplified genomic DNA showing correction of the c.A2987T mutation in the LQT2-hiPSCs[corr] line and mutation in the *NKX2.5*[eGFP/w] hESCs[N996I] line. The wild-type reference sequence (Ref) is shown in the top line. Source data for this figure is available on the online supplementary information page.

the end of the test pulse) and peak-tail $I_{Kr}$ density were decreased by 43 and 45%, respectively, corroborating the results obtained in the patient-specific hiPSC model. Importantly, while $I_{Kr}$ densities were affected by the mutation, voltage dependence of activation parameters (half-maximal voltage ($V_{1/2}$) and slope factor ($k$)) were similar, both in mutated and corrected LQT2-hiPSC-derived CMs and in wild-type and mutated hESC-derived CMs (Figure 6A and B). Furthermore, the time constant of $I_{Kr}$ activation ($\tau$) (Figure 6C and D), as well as the fast ($\tau_f$) and the slow ($\tau_s$) deactivation time constants were all unchanged when comparing wild-type and mutated cells (Figure 6E and F). Taken together, these findings suggest that the N996I HERG muta-

tion is the only relevant mutation that influences $I_{Kr}$ current in the heterozygous LQT2 CMs.

To evaluate the impact of the $I_{Kr}$ change on the APD, APs were recorded in single spontaneously contracting cells (Figure 7). Representative APs (paced at 1 Hz) from mutated and corrected hiPSC-derived CMs, and from wild-type and mutated hESC-derived CMs are shown in Figure 7A and C, respectively. The APD at 50 and 90% of repolarization (APD$_{50}$ and APD$_{90}$, respectively) was significantly reduced in the LQT2-hiPSC[corr] CMs compared with their mutated counterparts (Figure 7B). In the hESC model, only the APD$_{50}$ of *NKX2.5*[eGFP/w] hESC[N996I] CMs was significantly prolonged compared with the *NKX2.5*[eGFP/w] hESC CMs

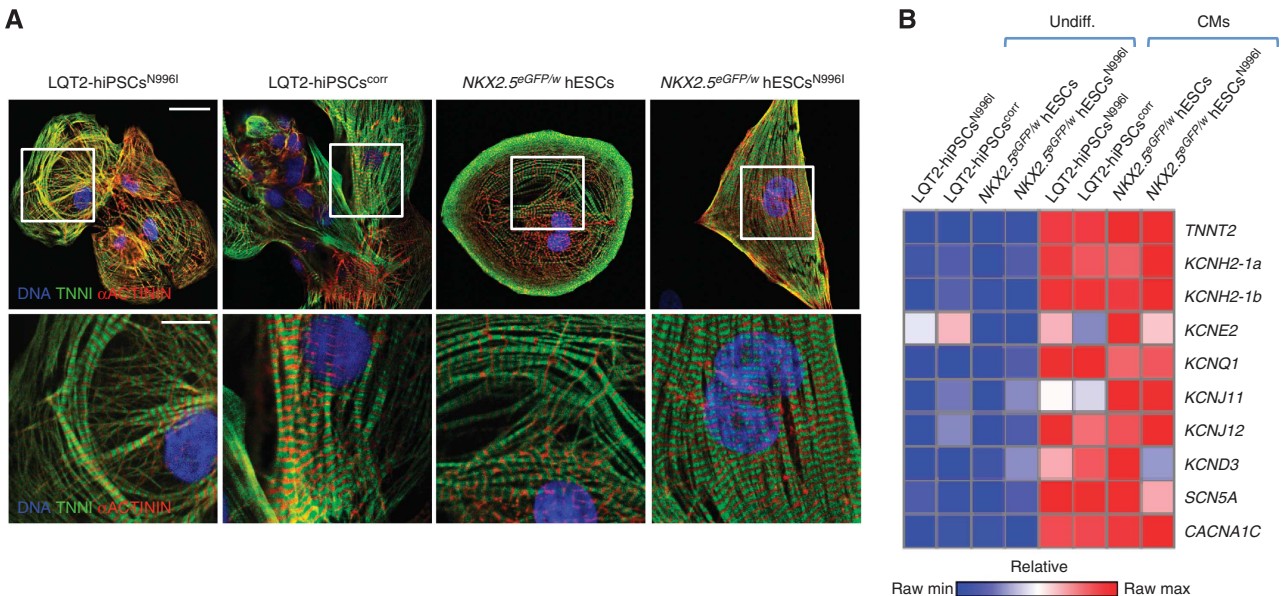

**Figure 3** Differentiation of hPSCs into the cardiac lineage. (**A**) Confocal immunofluorescence images of cardiac sarcomeric proteins TNNI (green) and α-actinin (red) in human CMs generated from LQT2-hiPSCs[N996I], LQT2-hiPSCs[corr], $NKX2.5^{eGFP/w}$ hESCs, and $NKX2.5^{eGFP/w}$ hESCs[N996I]. Nuclei are stained in blue. Bottom panels are a magnification of the area framed in the upper images. Top panels, scale bar: 25 μm; bottom panels, scale bar: 10 μm. (**B**) Transcriptional profile of human CMs generated from mutated and corrected hiPSCs (LQT2-hiPSCs[N996I] and LQT2-hiPSCs[corr], respectively) and from wild-type and mutated hESCs ($NKX2.5^{eGFP/w}$ hESCs and $NKX2.5^{eGFP/w}$ hESCs[N996I], respectively). Undifferentiated cells from each hPSC line are also shown (Undiff.). Quantitative RT–PCR analysis was performed on the cardiac troponin gene (*TNNT2*), to show enrichment for the cardiomyocyte population, on the HERG channel gene (*KCNH2*), and on other key genes encoding for ion channels involved in the generation of the action potential in cardiac cells. All values are normalized to *GAPDH* and are relative to undifferentiated $NKX2.5^{eGFP/w}$ hESCs. Raw minimum (min) and raw maximum (max) values were taken as a reference for heatmap representation.

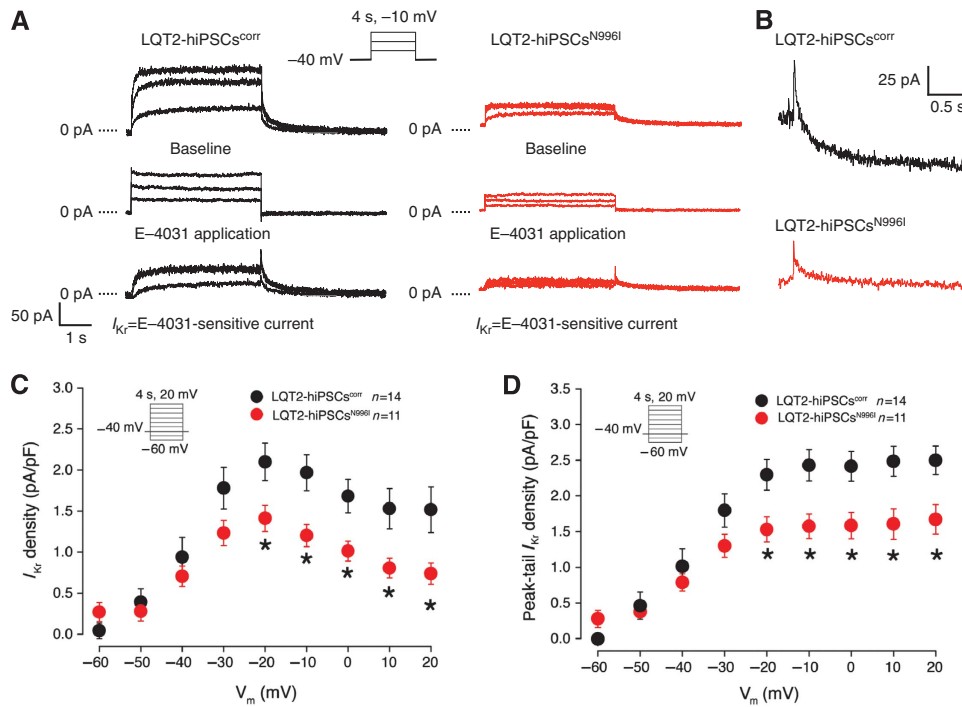

**Figure 4** $I_{Kr}$ densities in mutated and corrected LQT2-hiPSC-derived CMs. (**A**) Representative current traces elicited upon 4 s depolarizing voltage steps to −30, −20, and −10 mV from a holding potential of −40 mV, before and after the application of 1 μM E-4031. Inset: voltage protocol. (**B**) Typical tail currents measured after the depolarizing step to −10 mV in corrected (black) and mutated (red) LQT2-hiPSC-CMs, showing a bi-exponential decay. (**C**) Average current–voltage (*I–V*) relationships for $I_{Kr}$, measured at the end of the test pulses, in mutated (red) and corrected (black) LQT2-hiPSC-derived CMs. Inset: voltage protocol. * indicates statistical significance ($P = 0.046$, two-way rmANOVA; Holm–Sidak test *post hoc* analysis: −20 mV: $P = 0.015$, −10 mV: $P = 0.007$, 0 mV: $P = 0.018$, 10 mV: $P = 0.011$, 20 mV: $P = 0.013$). (**D**) Average *I–V* relationships for peak tail currents in mutated (red) and corrected (black) LQT2-hiPSC-derived CMs. * indicates statistical significance ($P = 0.039$, two-way rmANOVA; Holm–Sidak test *post-hoc* analysis: −20 mV: $P = 0.008$, −10 mV: $P = 0.004$, 0 mV: $P = 0.005$, 10 mV: $P = 0.004$, 20 mV: $P = 0.006$). Inset: voltage protocol.

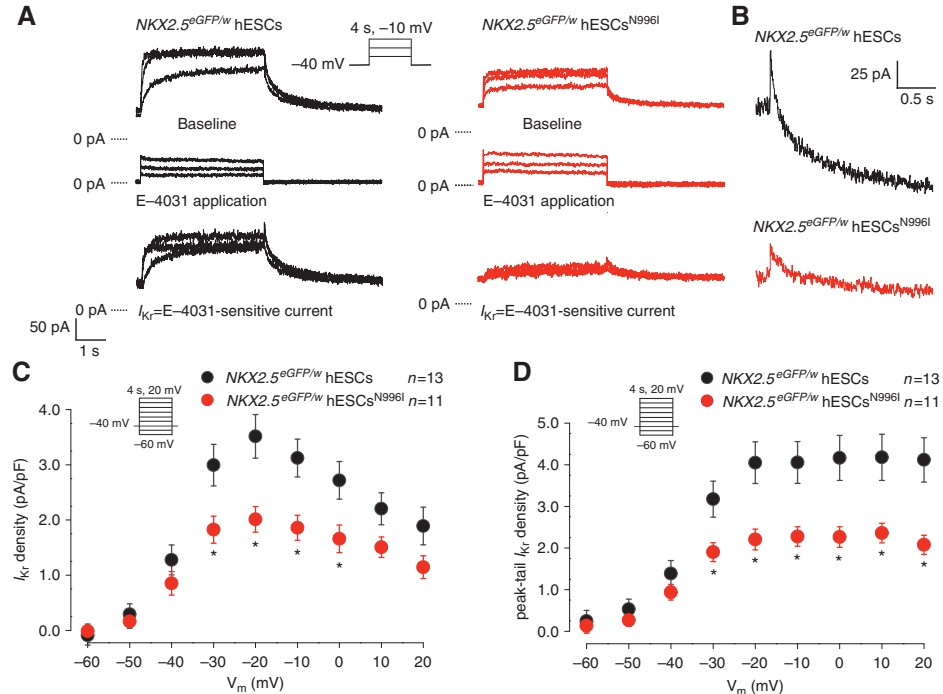

**Figure 5** $I_{Kr}$ densities in wild-type and mutated hESC-derived CMs. (**A**) Representative current traces elicited upon 4 s depolarizing voltage steps to $-30$, $-20$, and $-10$ mV from a holding potential of $-40$ mV, before and after the application of 1 µM E-4031. Inset: voltage protocol. (**B**) Typical tail currents measured after the depolarizing step to $-10$ mV in wild-type (black) and mutated (red) hESC-CMs, showing a bi-exponential decay. (**C**) Average $I$–$V$ relationships for $I_{Kr}$, measured at the end of the test pulses, in wild-type (black) and mutated (red) hESC-derived CMs. Inset: voltage protocol. * indicates statistical significance ($P = 0.026$, two-way rmANOVA; Holm–SIdak test *post hoc* analysis: $-30$ mV: $P = 0.004$; $-20$ mV: $P = 0.000$, $-10$ mV: $P = 0.002$, 0 mV: $P = 0.009$). (**D**) Average $I$–$V$ relationships for peak tail currents in wild-type (black) and mutated (red) hESC-derived CMs. * indicates statistical significance ($P = 0.014$, two-way rmANOVA; Holm–Sidak test *post-hoc* analysis: $-30$ mV: $P = 0.022$, $-20$ mV: $P = 0.001$, $-10$ mV: $P = 0.002$, 0 mV: $P = 0.001$, 10 mV: $P = 0.001$, 20 mV: $P = 0.003$). Inset: voltage protocol.

(Figure 7D). In contrast, in both the hiPSC and hESC paradigms, no differences in upstroke velocity ($V_{max}$), AP amplitude (APA), and maximal diastolic potential (MDP) were detected between the mutated and corrected patient lines, and between the wild-type and mutated hESC groups (Figure 7B and D, respectively).

### The c.A2987T (N996I) KCNH2 mutation causes a trafficking defect in the HERG channel

To investigate the functional consequences of the N996I-HERG mutation further, we examined the cellular distribution of the HERG channel in hiPSC- and hESC-derived CMs. Immunocytochemistry in both hPSC-CMs revealed a distribution of the HERG channel over the cell surface (Figure 8A), as well as in the endoplasmic reticulum (ER) and in the Golgi apparatus (Supplementary Figures S8 and S9). However, it was not possible to determine accurately by immunofluorescence whether CMs generated from the LQT2-hPSCs$^{N996I}$ models presented any differences in subcellular localization of HERG channels when compared to their wild-type or corrected counterparts. Therefore, we exploited the fact that we had introduced the c.A2987T (N996I) *KCNH2* mutation into an *NKX2.5*$^{eGFP/w}$ hESC line in which GFP reports the expression of the cardiac associated transcription factor NKX2.5 (Elliott *et al*, 2011). This allowed us to purify a population enriched for CMs by flow cytometry (Figure 8B), and quantify HERG channel levels within this fraction by western blot analysis. Two HERG protein bands were identified consistent with core glycosylated (135 kDa, immature) and more complex glycosylated (155 kDa, mature) isoforms

(Figure 8C), with the latter present predominantly in the *NKX2.5*-eGFP$^+$ fraction. In the *NKX2.5*-eGFP$^+$ hESC$^{N996I}$ cells, the 155-kDa protein band, representing the form transported to the cell membrane through the Golgi, was reduced by $\sim$2-fold compared to the wild-type cells, while the 135-kDa band, which corresponds to the protein located in the ER, was unaffected (Figure 8D). This resulted in a significant decrease in HERG trafficking efficiency, calculated as a ratio of the fully glycosylated 155 kDa band over total HERG protein (155 kDa band + 135 kDa band), of $\sim$40% (Supplementary Figure S10A). Similar analysis in LQT2-hiPSC$^{corr}$ and LQT2-hiPSC$^{N996I}$ CMs confirmed a specific reduction in the complex glycosylated HERG protein in the mutated cells compared to the corrected counterparts (Figure 8E and F; Supplementary Figure S10B). Thus, the N996I mutation appears to interfere with the maturation and trafficking of the channel to the membrane.

To further investigate the pathological mechanism of the N996I mutation, we tested whether this mutation resulted in a misfolded HERG protein by analysing the unfolded protein response (UPR) in CMs purified from *NKX2.5*$^{eGFP/w}$ hESCs and *NKX2.5*$^{eGFP/w}$ hESCs$^{N996I}$. UPR is an ER stress pathway that increases the synthesis of chaperones, such as Calnexin and Calreticulin, which in turn target misfolded proteins for proteasome degradation (Ellgaard and Helenius, 2003; Hetz, 2012). Western blot for the Activating Transcription Factor 6 (ATF6), a key regulator of transcriptional control in the mammalian UPR (Haze *et al*, 1999; Chen *et al*, 2002), revealed the presence of several forms of ATF6, including both the unprocessed protein embedded in the ER (90 kDa

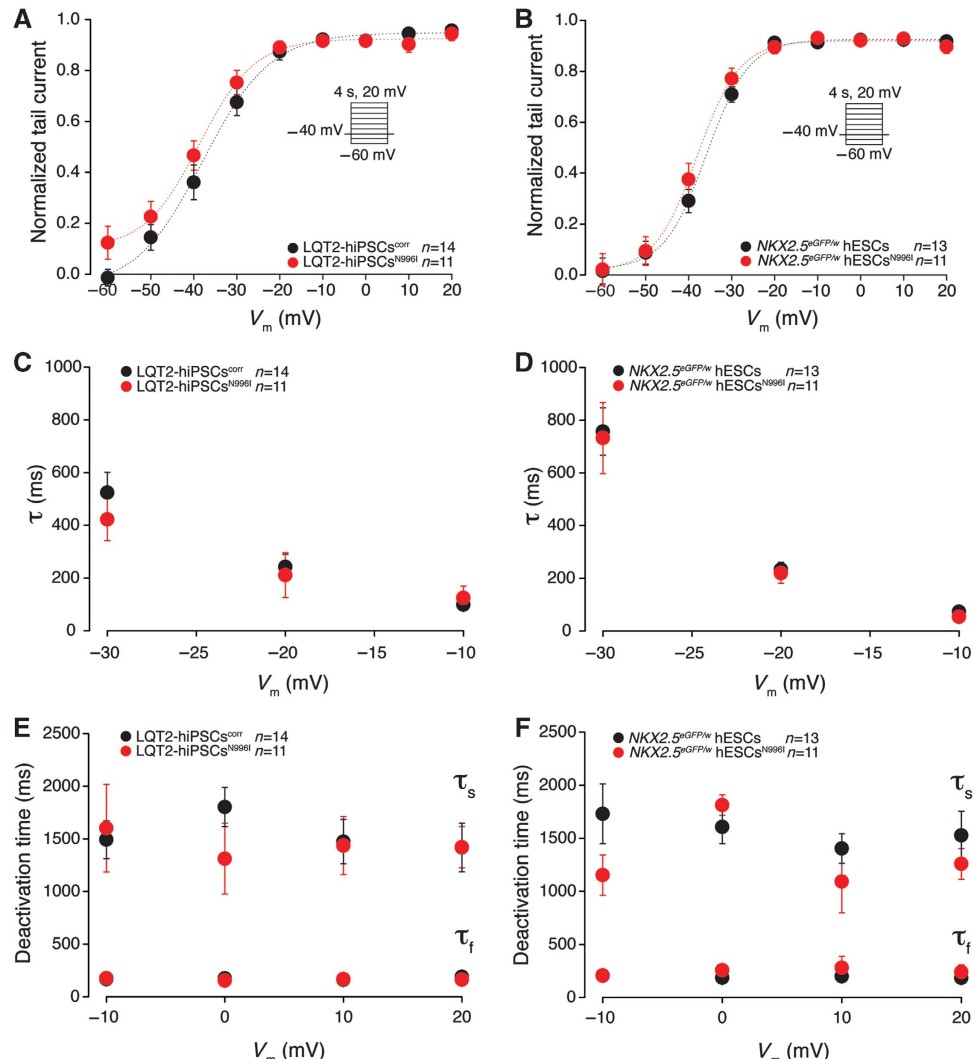

**Figure 6** $I_{Kr}$ activation and deactivation properties in mutated and corrected LQT2-hiPSC-derived CMs and in wild-type and mutated hESC-derived CMs. (**A**, **B**) Average peak tail current normalized to the maximal current following repolarization to $-40$ mV in mutated (red) and corrected (black) LQT2-hiPSC-derived CMs (**A**) and in wild-type (black) and mutated (red) hESC-derived CMs (**B**). Inset: voltage protocol. (**C**, **D**) Average time constants of $I_{Kr}$ activation ($\tau$) in mutated (red) and corrected (black) LQT2-hiPSC-derived CMs (**C**) and in wild-type (black) and mutated (red) hESC-derived CMs (**D**). (**E**, **F**) Average slow and fast time constants of $I_{Kr}$ deactivation ($\tau_s$ and $\tau_f$, respectively) in mutated (red) and corrected (black) LQT2-hiPSC-derived CMs (**E**) and in wild-type (black) and mutated (red) hESC-derived CMs (**F**).

band) and some smaller forms resulting from the cleavage of the full-length ATF6 during ER stress, with no detectable differences between wild-type and mutated *NKX2.5*-eGFP[+] cells (Figure 9A). Moreover, there was no evidence of an increase in the expression of the chaperones Calnexin and Calreticulin in *NKX2.5*[*eGFP/w*] hESC[N996I] CMs (Figure 9B and C), indicating that the N996I HERG mutation does not appear to stimulate the UPR pathway. Nevertheless, to assess the involvement of proteasome-mediated and/or lysosome-mediated degradation of HERG channel further, we determined the effects of the proteasome inhibitor lactacystin and the lysosome inhibitor leupeptin on the steady-state HERG protein levels in wild-type and mutated hESC-CMs. Following purification of the *NKX2.5*-eGFP[+] cell population by flow cytometry, cells were treated for 24 h with either lactacystin (20 μM) or leupeptin (100 μM) and cell lysates were then immunoblotted with anti-HERG antibody (Figure 9D). Treatment with the proteasome inhibitor lactacystin augmented the level of the mature, highly glycosylated form of HERG

in *NKX2.5*[*eGFP/w*] hESC[N996I] eGFP[+] cells, resulting in an ∼1.4 fold increase in channel trafficking efficiency. No increase was observed in the wild-type counterparts or after treatment with the lysosome inhibitor leupeptin (Figure 9D and E), suggesting a role of proteasomes in the degradation of N996I HERG protein.

## Discussion

The ability to reprogramme somatic cells from any individual of choice has created unprecedented opportunities to study disease development *in vitro* and create novel platforms for drug screening and discovery. However, one crucial limitation for disease modelling in hiPSCs has been that, unlike experimental animal models, humans are genetically heterogeneous (Goldstein, 2009; McKernan *et al*, 2009), and genetically matched controls are necessary to distinguish disease-relevant phenotypic changes from normal background-related variations due to polymorphisms in

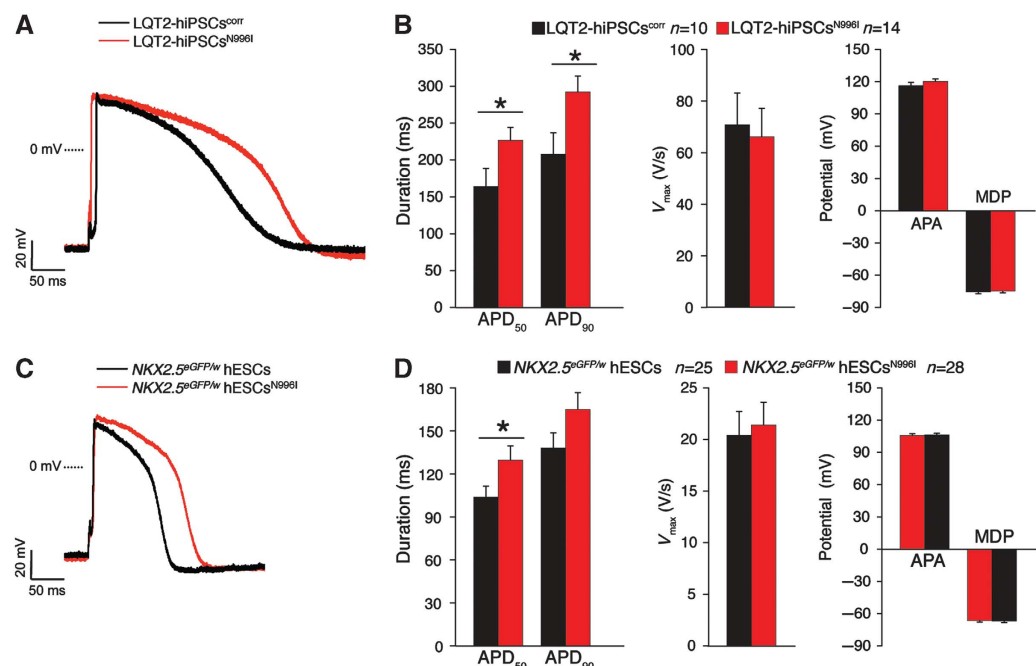

**Figure 7** AP characteristics in mutated and corrected LQT2-hiPSC-derived CMs and in wild-type and mutated hESC-derived CMs. (**A**, **B**) Representative examples of AP measured at 1 Hz (**A**) and average $APD_{50}$, $APD_{90}$, $V_{max}$, APA, and MDP (**B**) in mutated (red) and corrected (black) LQT2-hiPSC-derived CMs. * indicates statistical significance ($APD_{50}$: $P=0.045$, $APD_{90}$: $P=0.026$; *t*-test). (**C**, **D**) Representative examples of AP measured at 1 Hz (**C**) and average $APD_{50}$, $APD_{90}$, $V_{max}$, APA, and MDP (**D**) in wild-type (black) and mutated (red) hESC-derived CMs. * indicates statistical significance ($APD_{50}$: $P=0.049$, *t*-test).

other genes. Here, we generated a panel of isogenic control and cardiac disease lines and investigated the disease phenotype in parallel in both hiPSCs and hESCs harbouring the same N996I *KCNH2* mutation. Although this mutation has been described as a novel LQT2 variant and included in studies exploring possible genotype–phenotype correlations (Khositseth *et al*, 2004; Tester *et al*, 2005; Tan *et al*, 2006), its mechanism of action has remained unclear. Here, we demonstrated that this specific mutation is the cause of the LQT2 phenotype observed in the hPSC-derived CMs. Furthermore, we investigated the biophysical disease mechanism in functionally relevant human cardiac cells. Our findings demonstrated that the N996I *KCNH2* mutation caused a similar ($\sim 30$–$40\%$) $I_{Kr}$ reduction in both mutated hiPSC- and hESC-derived CMs. In particular, because a decrease of $\leqslant 50\%$ in the cardiac repolarizing potassium current indicates haploinsufficiency (Moss *et al*, 2007; Vandenberg *et al*, 2012), our results suggested this as the mechanism underlying $I_{Kr}$ reduction, rather than a dominant-negative effect of mutant subunits on the wild-type subunits. Furthermore, the reduction in the fully glycosylated mature 155 kDa HERG protein in the mutated hPSC-CMs and its partial rescue after treatment with the proteasome inhibitor lactacystin suggest that the mutation might induce, to some extent, protein clearance through the proteasomes, thus resulting in defective channel trafficking (Vandenberg *et al*, 2012). Further analysis of HERG protein stability and internalization is required to unravel the additional mechanisms through which mutated N996I HERG channels may be processed. It is noteworthy that, in contrast to previous reports examining trafficking-deficient HERG mutants in heterologous systems (Smith *et al*, 2011; Guo *et al*, 2012; Wang *et al*, 2012), we did not observe any

accumulation of the core-glycosylated form of HERG in the ER nor an activation of the UPR in the mutated hPSC-derived CMs. This might reflect either the specific effect of this particular mutation or the different cell systems used to investigate the pathophysiological mechanism. In overexpression systems, the levels of both wild-type and mutated HERG proteins are far from physiological; this could alter molecular stoichiometry and cause a more pronounced stimulation of ER stress and clearance pathways than in the native conditions of a CM.

It is widely recognized that hPSC-derived CMs are immature, with electrical properties, gene expression and contraction forces only equivalent to those of fetal or neonatal human myocytes (de Boer *et al*, 2010; Hoekstra *et al*, 2012). Nevertheless, our results show that these cells are capable of capturing specific traits of an electrical disease of the heart, such as LQT2, corroborating earlier findings (Itzhaki *et al*, 2011; Matsa *et al*, 2011). Of note, AP parameters (especially upstroke velocity) suggested that hESC-derived CMs were electrophysiologically less mature than those derived from the hiPSCs used here. While this did not preclude detection of $I_{Kr}$ reduction as a specific effect of the N996I-HERG mutation, the electrophysiological immaturity of the hESC-CMs might have influenced the impact of $I_{Kr}$ reduction on APD, where we observed a significant prolongation of $APD_{50}$ but not of both $APD_{50}$ and $APD_{90}$ as observed in hiPSC-CMs. In this regard, the longer plateau phase of the AP in hiPSC-CMs compared to hESC-CMs would allow a higher number of HERG channels to inactivate, resulting in a different contribution of $I_{Kr}$ to AP repolarization (Rudy, 2008). These differences could also be due to inherent variability among individual hiPSC and hESC lines (Narsinh *et al*, 2011; Mummery *et al*, 2012) but do not detract from the

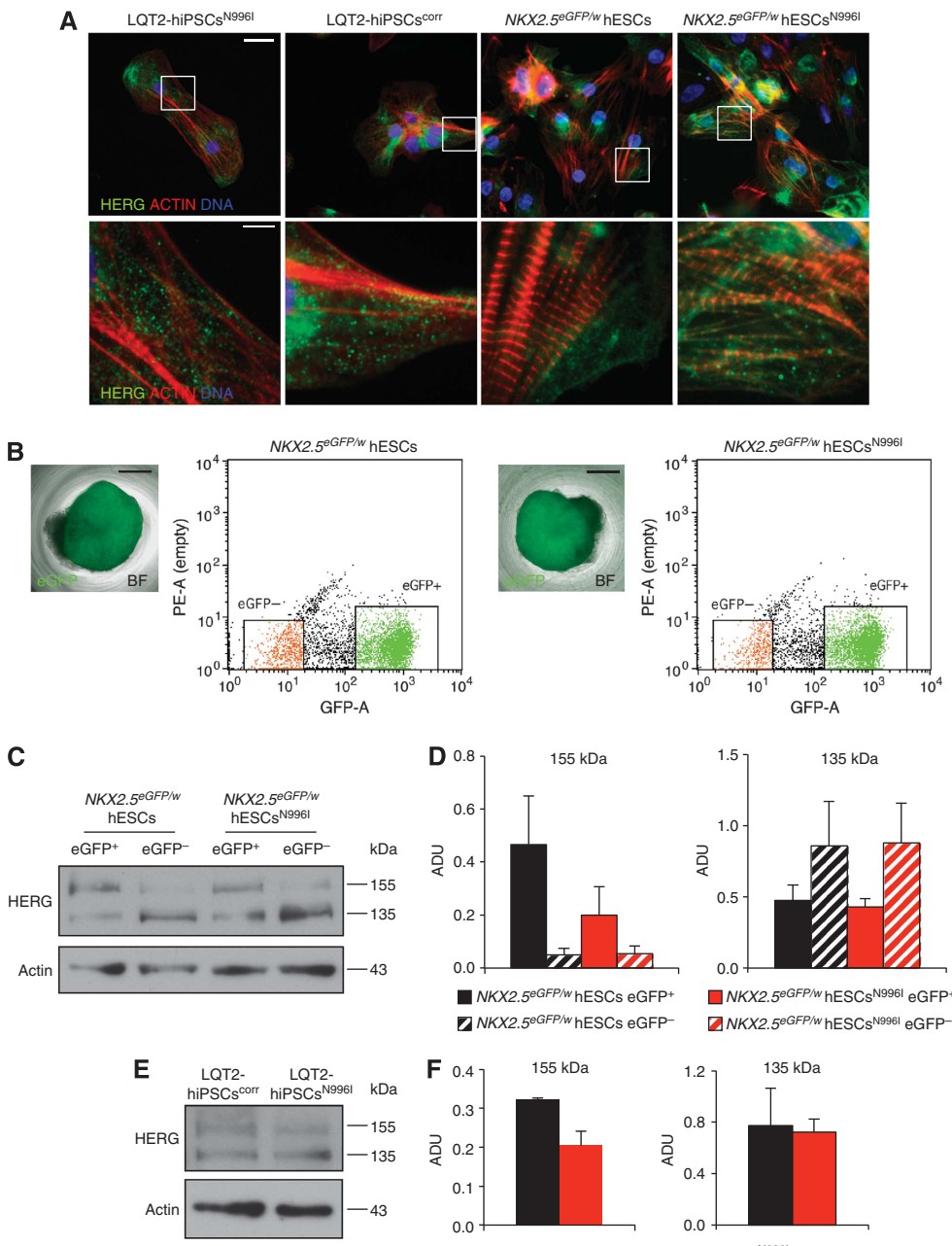

**Figure 8** Trafficking defect in CMs harbouring the c.A2987T (N996I) *KCNH2* mutation. (**A**) Immunofluorescence images of HERG channel (green) and actin (red) in representative human CMs derived from mutated and corrected LQT2-hiPSCs, and from wild-type and mutated hESCs. Nuclei are stained in blue. Bottom panels are a magnification of the area framed in the upper corresponding images. Top panels, scale bar: 25 µm; bottom panels, scale bar: 5 µm. (**B**) Flow cytometry purification of NKX2.5 eGFP$^+$ and NKX2.5 eGFP$^-$ hESC population from embryoid bodies differentiated from wild-type (left) and mutated (right) hESCs. The pictures show an individual eGFP-expressing (green) embryoid body; BF: bright field; scale bars: 400 µm. The representative dot plots show flow cytometric isolation of eGFP$^+$ (green) and eGFP$^-$ (orange) cell populations. (**C**) Representative western blot analysis of HERG protein in eGFP$^+$ and eGFP$^-$ cell populations purified from differentiated wild-type and mutated hESCs. Core- and complex-glycosylated HERG (135 and 155 kDa, respectively) are indicated. Actin is shown as a loading control. (**D**) Densitometric quantification of the 155-kDa and 135-kDa bands corresponding to the complex- and core-glycosylated HERG channel, respectively; ADU: arbitrary densitometric units; values are presented as mean ± s.e.m., $n = 4$. (**E**) Representative western blot analysis of HERG protein in LQT2-hiPSCs$^{corr}$- and LQT2-hiPSCs$^{N996I}$-derived CMs. Core- and complex-glycosylated HERG (135 and 155 kDa, respectively) are indicated. Actin is shown as a loading control. (**F**) Densitometric quantification of the 155-kDa and 135-kDa bands corresponding to the complex- and core-glycosylated HERG channel, respectively; ADU: arbitrary densitometric units; values are presented as mean ± s.e.m., $n = 2$. Source data for this figure is available on the online supplementary information page.

conclusion since isogenic pairs were always compared. In addition, while the $I_{Kr}$ traces recorded in our lines are in agreement with $I_{Kr}$ previously reported in hPSC-derived CMs (Moretti *et al*, 2010; Itzhaki *et al*, 2011; Ma *et al*, 2011; Lahti *et al*, 2012), the current density measured in our hESC-CMs was higher (3.5 pA/pF) than those described in the literature for both hiPSC-derived (0.55–1.9 p/pF) and native human (0.25–0.6 pA/pF) CMs (Hoekstra *et al*, 2012). The molecular basis of this discrepancy requires further independent investigation, but a larger $I_{Kr}$ could impact on the AP

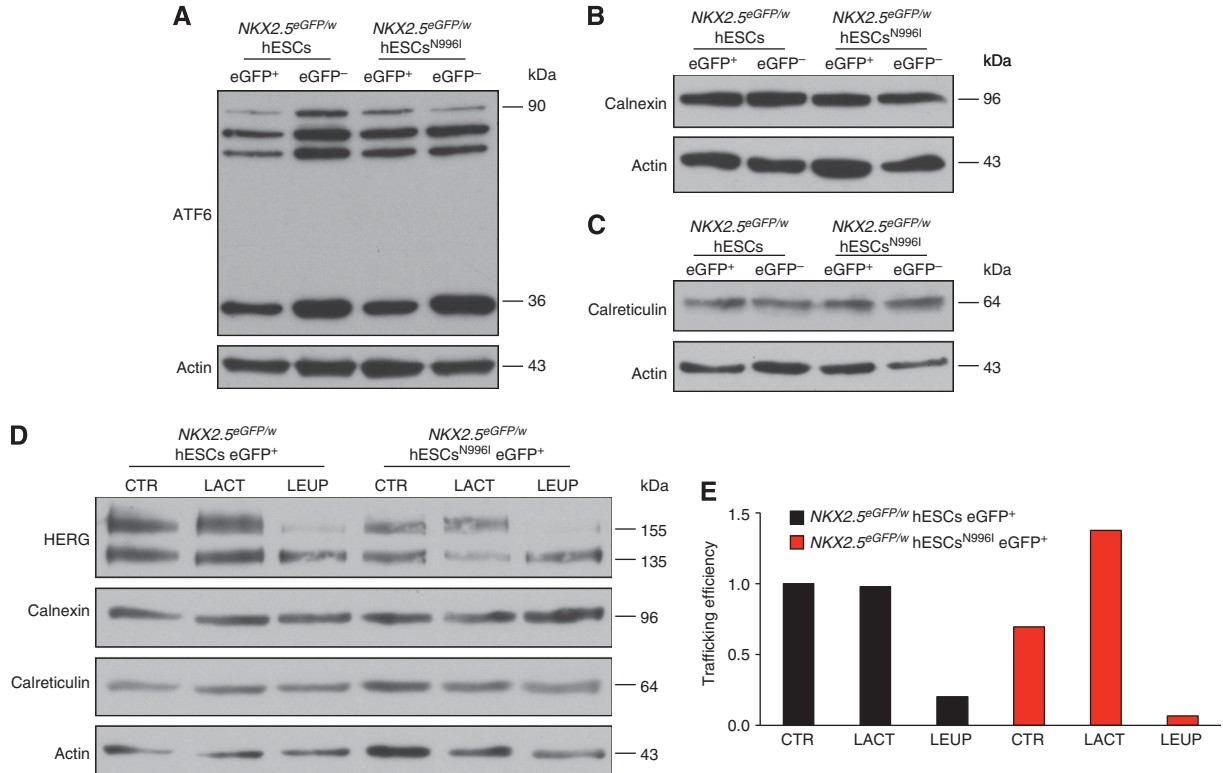

**Figure 9** UPR pathway analysis in wild-type and mutated hESC-CMs. (**A–C**) Western blot analysis of ATF6 (**A**), Calnexin (**B**), and Calreticulin (**C**) in eGFP$^+$ and eGFP$^-$ cell populations purified from differentiated wild-type and mutated hESCs. Actin is shown as a loading control. (**D**, **E**) Western blot analysis of HERG channel, calnexin, and calreticulin (**D**) and quantification of trafficking efficiency (**E**) under basal conditions (CTR) and upon proteasome (LACT) or lysosome (LEUP) inhibition in eGFP$^+$ cell populations purified from differentiated wild-type and mutated hESCs. Actin is shown as a loading control. Trafficking efficiency = fg/(fg + cg), where fg = fully-glycosylated 155 kDa band and cg = core-glycosylated 135 kDa band. Source data for this figure is available on the online supplementary information page.

characteristics, for example, contributing to a faster repolarization resulting in a shorter AP. Indeed, while the APD values measured in the LQT2-hiPSC$^{corr}$ CMs fall into the range reported for hPSC-derived CMs (Hoekstra *et al*, 2012), in the *NKX2.5$^{eGFP/w}$* hESC CMs the APD is rather small.

Another aspect of our work is noteworthy: we measured very modest APD prolongation in both LQT2-hiPSC$^{N996I}$ and *NKX2.5$^{eGFP/w}$* hESC$^{N996I}$ CMs when compared to their wild-type counterparts, with values shorter than those reported in other LQT2 hiPSC models (Itzhaki *et al*, 2011; Matsa *et al*, 2011; Lahti *et al*, 2012), and did not observe any early-after depolarization in the mutated cells. These differences are likely due to the specific N996I mutation, which results in a less severe LQT2 phenotype than those caused by other trafficking-deficient mutations (Moss *et al*, 2002). However, it should be noted that in all previous studies using LQT2 hiPSC models the control group consisted of wild-type hiPSCs derived from a healthy individual unrelated to the LQT2 patient, therefore over-estimation of differences cannot be excluded under these circumstances. The advantage of using isogenic lines is indeed highlighted by our results, where even subtle differences could be appreciated.

Most importantly, our data provide evidence that the pathogenesis of the N996I-HERG mutation can be modelled exclusively in CMs generated from hPSCs, without the need for heterologous and/or overexpression systems. In particular, we detected and analysed physiological levels of HERG protein (by immunostaining and western blot), as well as molecular chaperones and transcriptional regulators of

the UPR, in functionally relevant human cardiac cells. Moreover, the use of an *NKX2.5$^{eGFP/w}$* hESC reporter line allowed selection of the cardiac cell population from other cell types, such as neurons, endocrine cells, and undifferentiated proliferating cells, that also express *KCNH2* (Rosati *et al*, 2000; Pillozzi *et al*, 2002; Huffaker *et al*, 2009; Vandenberg *et al*, 2012). This highlights the particular value of combining patient-specific iPSC technology and genetic modification tools.

Since our study compared genetically matched lines differing uniquely in one LQT2 point mutation, a clear genotype–phenotype relationship could be ascertained. This is of particular value in avoiding over- or under-estimation of the impact of disease-related changes as a result of inadvertent bias when choosing a random control (even if age and gender matched with the same reprogramming and differentiation method). Therefore, this approach is a powerful means of identifying links between specific genotypes and cardiac disease predisposition, as also recently demonstrated for neurodegenerative disease (Reinhardt *et al*, 2013).

Finally, targeted gene correction of *KCNH2* in hiPSCs rescued $I_{Kr}$ density in hiPSC-CMs and normalized AP values. Although precise gene modification of human CMs *in situ* is not an immediate option for regenerative medicine or for treating LQTS, gene targeting as carried out here provides essential tools for deciphering molecular mechanisms, studying genetic–phenotypic interaction and perhaps, in the future, analysing the potential disease-causing SNPs identified by genome-wide association studies in humans. Other rescue approaches, such

as RNA interference, may only result in partial knock down of the mutated mRNA (Matsa *et al*, 2013), while genetic correction should restore the normal genetic phenotype in monogenic diseases. This can then demonstrate whether the specific genetic lesion is pathogenic or whether there are likely to be other modifier loci involved. Considering the increasing awareness of the broad influence of genetic background (Montgomery *et al*, 2010) and the biological differences between hPSC lines (Boulting *et al*, 2011), our experimental system may overcome some of the shortcomings of conventional hiPSC approaches in identifying disease-related phenotypes. As more LQTS-hiPSC lines are investigated using this approach, the molecular and genetic reasons for variability of expression and for incomplete penetrance seen in this, and other channelopathies, may be elucidated.

In summary, we have demonstrated that genetic correction of the N996I KCNH2 mutation associated with LQT2 restores $I_{Kr}$ density and normalizes APD in patient-specific LQT2-hiPSC-CMs. The same mutation introduced into a different genetic background (hESC) results in a very similar electrophysiological phenotype. Furthermore, the reduction in the mature complex-glycosylated 155 kDa protein band in the mutated hPSC-CMs suggests that, due to the mutation, a smaller number of channels are present at the cell membrane. This could be partially due to proteasome targeting for degradation, but might not be the only mechanism for this N996I-specific mutation. Further studies are required to assess more in detail the exact contribution of proteasome clearance pathway, protein channel stability, and lysosome degradation. Our study demonstrates that isogenic pairs of hPSCs can be used (1) to prove the authenticity of the genotype–phenotype correlation and (2) to unravel the pathophysiological mechanism seen in a genetically inherited cardiac disease. As such, this approach represents a robust strategy to study not only cardiac disease mechanisms but also other genetic disorders.

# Materials and methods

### Clinical history and genetic phenotype
A 38-year-old Caucasian woman presented with a prolonged QT interval (QT interval corrected for heart rate (QTc) 617 ms) at the ECG and was diagnosed with type-2 LQTS. Genetic screening indicated a heterozygous mutation c.A2987T in exon 13 of the KCNH2 gene, resulting in the N996I missense mutation. The patient has thus far been asymptomatic. She is now treated with β-blockers.

### Generation of patient-specific hiPSCs, differentiation into CMs, and drug treatment
We recruited a 38-year-old Caucasian female LQT2 patient for dermal biopsy after obtaining written informed consent. Reprogramming of primary skin fibroblasts was performed as described previously (Moretti *et al*, 2010; Jung *et al*, 2012). Briefly, fibroblasts were infected with retroviruses encoding OCT4, SOX2, KLF4, and MYC and cultured on mouse embryonic feeder (MEF) cells until hiPSC colonies could be picked. Both hiPSCs and hESCs were maintained in culture using standard procedures (Takahashi *et al*, 2007a) and were differentiated using standard cardiogenic protocols (Mummery *et al*, 2003; Elliott *et al*, 2011). Spontaneously contracting areas were manually dissected and cultured further until days 20–60 of differentiation. Cells for physiological experiments were enzymatically dissociated into single cells (Moretti *et al*, 2010), plated on fibronectin-coated glass coverslips, and analysed within 7–15 days.

For drug treatment, *NKX2.5*-eGFP $^+$ cell populations were purified by flow cytometry and maintained in low attachment plates while treated for 24 h with 20 μM lactacystin (Sigma) or 100 μM leupeptin (Sigma); cells were then lysated and subjected to immunoblotting.

### Genomic sequencing
Genomic DNA was isolated from cultured cells using the Gentra PureGene Cell Kit (Qiagen). To confirm the presence or absence of the c.A2987T KCNH2 mutation in the hPSCs, and to exclude the presence of other variants in the KCNH2 locus, the coding region and the exon–intron boundaries were PCR amplified from genomic DNA (Supplementary Table S1). The PCR products were purified using the QIAquick PCR Purification kit (Qiagen) and sequenced.

### Generation and identification of targeted hiPSCs and hESCs
A BAC (RP11-10L20) containing the ∼33-kb human KCNH2 locus was modified stepwise by recombineering to generate the final targeting vector (Fu *et al*, 2010). First, a loxP-flanked positive selection cassette (loxP-PGK-neo-loxP) comprising a mammalian promoter (pGK), a bacterial promoter (gb3), and a G418/kanamycin-resistance gene (neo) was amplified from an R6K plasmid and inserted 0.9 kb downstream of the mutation site. A 15.6-kb fragment, including a 6.6-kb 5′-homology arm, the inserted G418-resistance cassette, and a 4.3-kb 3′-homology arm, was then sub-cloned into a minimal vector (Gene Bridges), generating the wild-type KCNH2 targeting vector (KCNH2-A-loxP-pGK-Neo-loxP) used to correct the mutation in the LQT2-hiPSCs$^{N996I}$. For hESC targeting, the c.A2987T mutation was introduced into this targeting vector using the QuickChange XL Site-Directed Mutagenesis Kit (Stratagene) (KCNH2-T-loxP-pGK-Neo-loxP). The two vectors, wild type and mutated, were linearized with the restriction enzyme NotI before electroporation (Costa *et al*, 2007). Targeted clones were identified using a PCR-based screening strategy. Homologous integration of both the 5′ and 3′ homology arms, as well as correction or introduction of the c.A2987T KCNH2 mutation in the resulting clones were confirmed by sequencing. The loxP-flanked G418-resistance cassette was excised using Cre recombinase from both a corrected- and non-corrected hiPSC clone, and a mutated hESC clone, and the resulting lines subcloned as described previously (Davis *et al*, 2008). Clonal lines were screened for the loss of the neomycin-resistance cassette and for the absence of the Cre recombinase expression plasmid by PCR (Supplementary Table S1). These PCR products were also sequenced to confirm excision of the neomycin-resistance cassette. Karyotype analysis was performed using COBRA-FISH as described elsewhere (Szuhai and Tanke, 2006). Twenty metaphase spreads for each sample were analysed.

### Immunofluorescence and western blot analysis
Cells were fixed in 4% paraformaldehyde, permeabilised with phosphate buffer saline (PBS)/0.1% Triton X-100 (Sigma-Aldrich), and blocked with 10% FCS (Life Technologies). Samples were incubated overnight at 4°C with primary antibodies specific for the following: NANOG (goat polyclonal, R&D Systems), SSEA4 and TNNI (mouse monoclonal and rabbit polyclonal, respectively, both from Santa Cruz Biotechnology), α-ACTININ and HERG (mouse monoclonal and rabbit polyclonal, respectively, both from Sigma-Aldrich), PDI (mouse monoclonal, Abcam), and Golgin-97 (mouse monoclonal, Molecular Probes). Primary antibodies were detected with either Cy3- or Alexa-Fluor 488-conjugated antibodies. Nuclei were visualized with DAPI (Invitrogen) and F-actin with Phalloidin-Alexa-Fluor-594-conjugate (Invitrogen). Images were captured using either a Leica DMI6000-AF6000 fluorescence microscope or a Leica SP5 confocal laser-scanning microscope (both from Leica Microsystems).

Western blotting on whole-cell lysate of either eGFP $^+$ and eGFP $^-$ cells purified from differentiated *NKX2.5*$^{eGFP/w}$ hESCs and *NKX2.5*$^{eGFP/w}$ hESCs$^{N996I}$ or of manually microdissected beating areas from differentiated LQT2-hiPSCs$^{N996I}$ and LQT2-hiPSCs$^{corr}$ was performed with standard protocols (Kurien and Scofield, 2006) using 40–80 μg proteins and the following primary antibodies: HERG (mouse monoclonal, Enzo Life Sciences), ACTIN (mouse monoclonal, Millipore), ATF6 (mouse monoclonal, ActiveMotif), Calnexin and Calreticulin (both rabbit polyclonal, Abcam). Western blots were quantified using the ImageJ software (http://rsb.info.nih.gov/ij/). Whole film images are provided in Supplementary data.

### Gene expression analysis

For qRT–PCR, total RNA was purified using the RNeasy Mini Kit (Qiagen) according to the manufacturer's protocol. RNA samples were reverse transcribed using the SuperScript II First-Strand Synthesis kit (Invitrogen). Gene expression was assessed by qRT–PCR as described previously (Moretti *et al*, 2010). Gene expression levels were normalized to *GAPDH* and represented in a heatmap using the GENE-E software (http://www.broadinstitute.org/cancer/software/GENE-E/index.html). Primer sequences are provided in Supplementary Table S1.

For whole-genome microarray, total RNA was isolated from undifferentiated hPSCs using the NucleoSpin miRNA kit (Macherey-Nagel) and hybridized on Illumina HT12v4 microarrays. The raw microarray data were analysed with the PluriTest algorithm (http://www.pluritest.org; Muller *et al*, 2011).

### Electrophysiological characterization

*Data acquisition and analysis.* Recordings were performed on single CMs 7–15 days after cell dissociation. For $I_{Kr}$ measurements, CMs were identified on the basis of their typical morphology, while for AP measurements, spontaneously contracting cells that could be paced at the stimulation frequency of 1 Hz were selected. Data were collected from at least three independent differentiations per line. $I_{Kr}$ and APs were recorded with the ruptured and the perforated patch-clamp technique, respectively, using an Axopatch 200B amplifier (Molecular Devices). Voltage control and data acquisition of $I_{Kr}$ and APs were performed with pClamp10.1 (Axon Instruments). Membrane currents were analysed with Clampfit 10.1 (Axon Instruments), while analysis of APs was performed with the custom-made software. Patch pipettes had a tip resistance of $\sim 2\,M\Omega$ and series resistance (Rs) was compensated by 80%. $I_{Kr}$ and APs were filtered (2 and 5 kHz, respectively), and digitized (2 and 40 kHz, respectively). Potentials were corrected for liquid junction potentials (Barry and Lynch, 1991), which were calculated using Clampex 10.1 (Axon Instruments) and were equal to 15 and 17 mV for AP and $I_{Kr}$ measurements, respectively.

*Voltage-clamp experiments.* $I_{Kr}$ was recorded at 37°C using a pipette solution containing (mM): K-gluconate 125, KCl 20, $K_2$-ATP 5, HEPES 10, EGTA 10; pH 7.2 (KOH). The bath solution was Tyrode's solution containing (mM): NaCl 140, KCl 5.4, $CaCl_2$ 1.8, $MgCl_2$ 1.0, glucose 5.5, HEPES 5; pH 7.4 (NaOH). Nifedipine (5 µM) and JNJ-303 (1 µM) were added to the external solution to block the L-type calcium current ($I_{CaL}$) and the slow component of the rectifier potassium current ($I_{Ks}$), respectively. Currents were elicited using 4 s hyper- and depolarizing pulses from a holding potential of −40 mV, as indicated in the voltage protocols depicted in Figures 4 and 5. The cycle length of the voltage clamp steps was 10 s. $I_{Kr}$ was measured as a 1-µM E-4031-sensitive current, by subtraction of the current recorded before and after E-4031 application. $I_{Kr}$ densities were calculated by dividing current amplitude (pA), measured either at the end of the test pulses (Figures 4C and 5C) or at the peak of the tail current (Figures 4D and 5D), by cell membrane capacitance (pF). Cell membrane capacitance was measured by dividing the decay time constant of the capacitive transient in response to 5 mV depolarizing steps from −40 mV, by the Rs. Average cell capacitance for both hiPSC- and hESC-CM groups are shown in Supplementary Figure S11.

Activation curves, determined from peak tail current normalized to the maximal value and plotted against the test pulse voltage, were fitted with Boltzmann equation ($y = [1 + \exp\{(V - V_{1/2})/k\}]^{-1}$), where $V_{1/2}$ is the half-maximal voltage of activation, and $k$ is the slope factor.

Time constants of activation ($\tau$) were obtained by fitting $I_{Kr}$ during depolarizing pulses with a single-exponential function ($y = A_0 + A_1\exp[-t/\tau]$). Finally, the slow ($\tau_s$) and fast ($\tau_f$) time constants of deactivation were determined by fitting $I_{Kr}$ tail currents, at voltage steps positive to −10 mV, with a bi-exponential function ($y = A_0 + A_f\exp[-t/\tau_f] + A_s\exp[-t/\tau_s]$).

*Current-clamp experiments.* APs were recorded at 37°C in Tyrode's solution. The pipette solution contained (mM): K-gluconate 125, KCl 20, NaCl 5, amphotericin-B 0.22, HEPES 10; pH 7.2 (KOH). APs were elicited at the stimulation frequency of 1 Hz by 3 ms, 1.2 ×

threshold current pulses through the patch pipette. MDP, $V_{max}$, APA, and $APD_{50}$ and $APD_{90}$ were analysed. Data from 10 consecutive APs were averaged.

*MEA electrophysiology.* MEA standard measurements were performed in DMEM culture medium supplemented with 2% FCS as previously described (Braam *et al*, 2013). E-4031 was dissolved in DMSO at 10 mM and serial dilutions were made in culture medium.

### Statistical analysis

The SigmaStat 3.5 software was used for statistical analysis. Results are expressed as mean ± s.e.m. The software performed tests for normality and equality of variance on the data sets prior to application of a statistical test. Group comparisons were made using unpaired Student's *t*-test when both conditions were fulfilled or Mann–Whitney test when the conditions of normality and equal variance were not met. For repetitive measurements, two-way rmANOVA followed by a Holm–Sidak test for *post-hoc* analysis was used. $P < 0.05$ was considered as statistically significant.

### Ethics statement

The protocols for research involving human subjects and for stem-cell research were approved by the institutional review board and the committee charged with oversight of embryonic stem-cell research at the Technical University of Munich. Written informed consent was received from the participant prior to inclusion in the study. The studies on hESCs were performed exclusively in the Netherlands where the ethical committee for research at the LUMC provided approval for their use.

### Supplementary data

Supplementary data are available at *The EMBO Journal* Online (http://www.embojournal.org).

## Acknowledgements

We thank C Höhnke (Technical University of Munich) for obtaining the skin biopsy; K Takashi and S Yamanaka (Kyoto University) for providing viral vectors through Addgene; F Stewart (Technical University of Dresden) for providing the R6K plasmid; MJM van der Burg, K Szuhai, and H Tanke (Leiden University Medical Center) for karyotyping analysis; AO Verkerk (University of Amsterdam), M Birket (Leiden University Medical Center), D Sinnecker (Technical University of Munich) and R Dirschinger (Technical University of Munich) for comments on the manuscript; S Böhringer (Leiden University Medical Center) for statistical assistance. We are especially thankful to the type-2 long-QT syndrome patient for tissue donation. This work was supported by grants from the European Research Council, ERC 261053 (K-LL) and ERC 323182 (CLM); the German Research Foundation, Research Unit 923, Mo 2217/1-1 (AM), La 1238 3-1/4-1 (K-LL), and WE 1639/3-1 (AW); German Centre for Cardiovascular Research (K-LL and AM); ZoNMW Animal Alternatives, 114000101 (CLM, SC, and LT); Netherlands Proteomics Consortium, 050-040-250 (CLM); Netherlands Institute of Regenerative Medicine (CLM and RD); EU Marie Curie FP7-people-2011-IEF programme, HPSCLQT 29999 (MB) and a short-term EMBO fellowship, ASTF 387.00-2011 (CD'A).

*Author contributions*: MB, AM, CLM, and K-LL conceived the project and designed experimental details. MB performed hiPSC reprogramming together with CBJ. SC performed the single-cell electrophysiology measurements together with AW. LGJT did the MEA experiments. The targeting strategy was designed by RPD and MB, and experimental targeting work was done by MB and DW. DE provided the *NKX2.5*-hESC line. CD'A performed the FACS sort and western blot experiments. JH did the HERG immunofluorescence staining. The manuscript was written by MB, CLM, and AM.

## Conflict of interest

Christine Mummery is co-founder and advisor of Pluriomics. The remaining authors declare that they have no conflict of interest.

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
