## [Review Process File · The EMBO Journal]

Manuscript EMBO-2013-85550

Isogenic human pluripotent stem cell pairs reveal the role of a KCNH2 mutation in long-QT syndrome

Milena Bellin, Simona Casini, Richard P. Davis, Cristina D'Aniello, Jessica Haas, Dorien Ward-van Oostwaard, Leon G. J. Tertoolen, Christian B. Jung, David A. Elliott, Andrea Welling, Karl-Ludwig Laugwitz, Alessandra Moretti, and Christine L. Mummery

Corresponding author: Milena Bellin, Leiden University Medical Center

Review timeline:

Submission date:	03 May 2013
Editorial Decision:	03 June 2013
Revision received:	30 August 2013
Editorial Decision:	13 September 2013
Revision received:	07 October 2013
Accepted:	11 October 2013

Transaction Report:

Editor: Thomas Schwarz-Romond

1st Editorial Decision

03 June 2013

Thank you very much for submitting your paper on isogenic pairs of patient-derived cardiac myocytes to model the long QT syndrome for consideration to The EMBO Journal editorial office.

The attached comments reveal timeliness and potential clinical relevance of the study. Accordingly, ref#1 and #3 would be satisfied by significant though necessary methodological clarifications. In contrast and while appreciating the elegant isogenic approach, ref#2 does not recognize any significant novel molecular insights. As this definitely impinges on the more general advance associated with your study, we very much encourage you to further develop the mechanistic lead into HERG trafficking/internalization, ubiquitination and/or protein stability before a final version of your study could ultimately be considered for publication at our title.

Please do not hesitate to contact me in case further questions arise. (Given our time demands, preferably via E-mail).

I am very much looking forward to your revised paper and remain with best regards.

REFEREE REPORTS

Referee #1:

This paper describes an isogenic approach for modeling the long QT syndrome in patient-derived cardiac myocytes differentiated from stem cells. As the authors correctly point out, this is timely and represents an advance in the technique by eliminating problems caused by potential modifier loci, variabilities inherent in the iPSC procedure, and heterogeneities in the starting cell population. The manuscript is, on the whole, well written and comprehensive. A small number of omissions in detail require attention as follows:

(1) Major Concerns:

(a) Statistics: The EMBO Journal requires that the actual P value, name of statistical test and number of replicate experiments be quoted for every statistical comparison. As it stands, a blanket statement in Methods stating that "P<0.05 is taken as significant" is inadequate and contravenes the Instructions to Authors. Please, at least state the actual P value (preferably with name of test) for every statistical comparison made.

(b) Electrophysiology: In Methods, first section (p. 19) you state that potentials were corrected for an estimated liquid junction potential. It is essential that you state what this correction was (in mV), and how the estimation was made.

In Figure 4, IKr currents look uncharacteristic in that, there is little or no evidence for the upswing of tail current due to recovery from inactivation, prior to its exponential decay. There is no indication if this is due to the voltage you chose for displaying the current, nor do you state how you quantified IKr, which is normally measured at the tail current peak. I would like you to split Figure 4 into two (a total of 9 figures are permitted per article), with a new Figure addressing the hiPSC results and one for the hESC results, each in more detail. Please refer to Figures 1 - 2 in Sanguinetti et al. (Cell 81, p299-307) for the degree of characterization needed. It will reassure the reader that these cells generate an IKr that agrees with the literature, or if that is not the case, then please account why the current is non-classical. I would like to see IKr at two or three different hyperpolarization voltages to show the biphasic tail current, and for you to indicate the saturation of the so-called inactivating current as the depolarization voltage (first step of the two-step voltage protocol) is made more positive. It is informative to have this for both isogenic lines, hence my asking for two figures, particularly since Figure 5 shows no significant differences in any of the other measured parameters.

In Figure 6 and Supplementary Figure 7, all the action potentials and field potentials are unexpectedly short in duration compared with other published accounts (Itzhaki et al. Nature 471, p225-9; Doss et al. PLoS ONE 7, p1-17). Please account for this discrepancy. Moreover, early afterdepolarizations (EADs) were not present in the prolonged action potentials (at least this is not discussed). If EADs were absent and AP prolongation was modest, please address if this is related to the observed modest loss of IKr density (30-40%). Many class 2 hERG mutations show a much greater loss of function (Anderson et al. Circulation 113, p 365-373), please account for this additional discrepancy with published findings. Is it certainly possible that the c.A2987T KCNH2 mutation is less severe than the previously reported mutations, and this is a good rationale for using isogenic lines that can detect the result of more subtle mutations.

Finally, the action potentials were recorded using the perforated patch method. Please reassure the reader that this was monitored by stating all of the measures you took to be certain that patches remained perforated, and did not rupture.

(2) Minor comments

Introduction, paragraph 2, the word "afflicted" is anthropomorphic and should be replaced (affected?)

Results, section 3, paragraph 3 (page 9), where it states that 1 μ M E-4031 was used, the font looks odd. Is the number inadvertently subscripted here?

(3) Additional suggestions

In general, I found that the Introduction was too long and Discussion too short. I think that this manuscript might benefit from reversal of the relative lengths of these two sections. I think you might want to make the Introduction more terse, cutting out some superfluous details.

Referee #2:

The manuscript by Bellin et al. described providing human cellular model of long QT syndrome type 2 (LQTS2) to investigate molecular and cellular mechanisms underlying this disease. In this study, the authors generated induced pluripotent stem (iPS) cells from patient and corrected the mutation A2987T in KCNH2 gene encoding HERG channel to generate isogenic iPS cells as control, and then differentiated the iPS cells into cardiac cells in order to record IKr current and action potentials using electrophysiological tools, to examine gene expression using RT-PCR and to quantify glycosylated HERG proteins using Western blot. They found the mutation A2987T (N996I) in KCNH2 is the primary cause to prolong action potential in human cardiac cells in vitro. Also, they demonstrate reduction of membrane-localizing HERG channels in the patient cardiomyocytes. Overall, I feel the study timely to understand molecular basis of LQTS using a new cellular model. Their approach could also provide a useful platform to find new ways to investigate human-specific mechanisms of cardiac arrhythmias using patient-specific pluripotent stem cells, and it would be widely informative in the fields of cardiovascular study and interesting for audience in this journal. I believe this study is basically accomplished, but it has been known that some missense mutations in KCNH2 could reduce IKr current, resulting in prolonged action potential in iPS cell-derived cardiomyocytes. In this study, unfortunately, there are no new insights into the disease although the authors introduced the same point mutation in human embryonic stem cells, generating two genetically distinct isogenic pairs of LQTS and control lines. First of all, to confirm their findings, it would be great if the authors could address two main concerns below.

Major concerns:

1. In Figure 7C-D, the authors suggest that highly glycosylated HERG proteins (at 155 kDa) were reduced in cardiomyocytes expressing N996I mutant. However, the blot image of HERG protein doesn't seem to be much reliable. I believe there are some technical issues existing in the blotting steps (especially, transferring proteins in gel into PVDF membranes or incubation of the membrane to the antibodies) because 135 kDa HERG bands were not consistent among the samples (left, higher but right, lower). It'd be strongly recommended for the authors to repeat the experiments to show representative results as well as entire film image (not cutting and pasting only small area to demonstrate the bands), following recent regulation in Nature Publishing group (some journals has already asked the authors to provide whole film images as supplementary figures for all Western blots).
2. Although the authors mention that "Furthermore, the reduction of the mature complex-glycosylated 155 kDa protein band in the mutated hESC-CMs suggests that the mutation results in a misfolded protein that does not reach the Golgi and, most likely, the cell membrane", in this study the authors don't show any evidences to support this molecular phenotype. It would be required to examine if the proteins are aggregated in the ER/SR or the Golgi and if it could induce ER stress. Also, the mutation might affect some associating molecules that are essential for HERG trafficking/internalization, ubiquitination or protein stability. Providing novel insights into LQTS2 would be necessary rather than showing expected results such as prolonged action potential and reduced IKr current.

Minor points:

1. Combining Fig. 1 and 2 would be appropriate.
2. Teratoma formation assays would be required since in this study the gene expression profiling of in vitro spontaneous differentiation was very noisy. For instance, LQT2-hiPSCs-N996I showed 10-fold higher expression of PTF1A, CD31, MYL2 and TH genes than LQT2-hiPSC-corr although both Nkx2.5-GFP lines showed similar expressions.
3. In Figure 6A,C, there are some artifacts in depolarization phase of the action potentials.

Referee #3:

Bellin and colleagues present a manuscript examining a mutation in KCNH2 linked to the long QT syndrome using isogenic human pluripotent stem cells. Specifically, the authors study the A2987T (N996I) mutation in KCNH2 in isogenic induced pluripotent (iPS) cell lines and isogenic embryonic stem (ES) cell lines. The authors generated a patient-derived iPS cell line harboring the N966I mutation and performed genetic correction by homologous recombination to produce an isogenic line for comparison. In complementary studies, the authors introduced the N966I mutation into a wild type hESC reporter line ESC Nkx2.5eGFP. In both cases, the N966I mutation decreased I_{Kr} and prolonged action potential duration relative to the isogenic control. Furthermore, the authors provide evidence that the decreased current is due to a defect in trafficking of the mutant protein to the surface membrane. The strength of the study is that this is the first example to my knowledge using isogenic iPS and ES cell lines to demonstrate clearly and consistently a disease causing mutation in an inherited heart disease. There are some curiosities in the data such as the differences in the action potential properties comparing the iPS cell- and ES cell-derived cardiomyocytes. It seems that the ES cell-derived cardiomyocytes exhibit a more immature electrophysiological phenotype. Nevertheless, the N966I mutation prolongs the APD and decreases I_{Kr} in both cases. Overall, by overcoming the limitations associated with modifying genes and family controls, the authors provide clear results with the isogenic cell lines. The study is presented well, and the data are compelling to this reviewer. I have only minor technical questions as follow:

- 1)The methods state that electrophysiological studies were performed on cells 20-60 days in culture. This is quite a broad range, and other studies suggest that the cardiomyocytes exhibit a changing phenotype with some maturation over this time period. Does this range of differentiation time impact potentially on the differences observed between the iPS cells and ES cells?
- 2)In order to allow comparative studies in the future, it would be useful to know the average whole cell capacitances for the cells patched.
- 3) p. 7, line 5, "... resulting in an isoleucine to asparagine substitution at position 996 (N996I)..." should be "... resulting in an asparagine to isoleucine substitution at position 996 (N996I)..."

POINT-BY-POINT RESPONSE TO REFEREES

We thank the reviewers for their interest in our work, constructive criticisms and instructive comments. We have addressed each of the major issues they raised by carrying out an extensive set of new experiments, as well as *via* editorial revision. We believe these clarify key issues highlighted in their review. These changes have been incorporated into the revised version of the manuscript and Supplementary Information. The new manuscript text and Figures, as well as a point-by-point rebuttal are provided for the referees and editor.

Summary of the major changes: at the suggestion of Referee #1, we carried out detailed I_{Kr} analysis in both hiPSC- and hESC-CMs. This resulted in improved current characterisation, which is shown in a new Figure 4 (hiPSC I_{Kr}), a new Figure 5 (hESC I_{Kr}), and a revised Figure 6. These figures show I_{Kr} at three different depolarisation voltages, with a clearly visible peak tail, characteristic of I_{Kr} , and fast and slow time constants of deactivation at four depolarisation voltages. Furthermore, as requested by Referee #2, a new set of experiments addressed the expression level of HERG protein by Western blotting both in hiPSC- and hESC-derived cardiomyocytes. This resulted in a revised Figure 8. In addition, a completely new series of experiments addressed ER stress, and HERG protein accumulation and degradation and resulted in a new Figure 9. All the Western blotting assays presented in our study are now provided as whole-film images in the “Supplementary Information” section. Pluripotency of the hPSC lines used in our work was assessed using “PluriTest” and results are presented in the revised Supplementary Figure S6. As suggested by Referee #3, whole cell capacitance is now presented in the new Supplementary Figure S11.

We believe that our revised manuscript addresses all of the reviewers’ concerns as summarized in the point-by-point response below.

Referee #1:

This paper describes an isogenic approach for modeling the long QT syndrome in patient-derived cardiac myocytes differentiated from stem cells. As the authors correctly point out, this is timely and represents an advance in the technique by eliminating problems caused by potential modifier loci, variabilities inherent in the iPSC procedure, and heterogeneities in the starting cell population. The manuscript is, on the whole, well written and comprehensive. A small number of omissions in detail require attention as follows:

We thank the referee for appreciation of our work and approach for studying this KCNH2 mutation using isogenic pairs of human pluripotent stem cells.

(1) Major Concerns:

(a) Statistics: The EMBO Journal requires that the actual P value, name of statistical test and number of replicate experiments be quoted for every statistical comparison. As it stands, a blanket statement in Methods stating that "P<0.05 is taken as significant" is

inadequate and contravenes the Instructions to Authors. Please, at least state the actual P value (preferably with name of test) for every statistical comparison made.

We apologize for these omissions. As suggested by the referee, we have now indicated (in the Figure Legends) the actual P value and the name of the test for every statistical comparison (specifically in the new Figures 4 and 5 and in Figure 7). The number of replicates is specified in each Figure, where “n” indicates the number of measured cells. Data were collected from at least 3 independent differentiation experiments per line, as stated in the “Material and methods” section under “Electrophysiological characterization - Data acquisition and analysis”, page 22.

(b) Electrophysiology: In Methods, first section (p. 19) you state that potentials were corrected for an estimated liquid junction potential. It is essential that you state what this correction was (in mV), and how the estimation was made.

We have calculated the liquid junction potentials using Clampex (version 10.1) ‘Junction Potential Calculator’, which was originally based on the software by Professor Peter H. Barry. Using the pipette and extracellular solutions as mentioned in the “Material and methods”, we calculated liquid junction potentials of 15 (action potential measurements) and 17 (I_{Kr} measurements) mV. The software and the liquid junction potential values are now included in the revised version of the manuscript as well as a reference to the study of Barry and Lynch (1991) on liquid junction potentials (Barry & Lynch, 1991). The revised sentence (page 23) now reads: “Potentials were corrected for liquid junction potentials (Barry and Lynch, 1991), which were calculated using Clampex 10.1 (Axon Instruments) and were equal to 15 and 17 mV for AP and I_{Kr} measurements, respectively”.

In Figure 4, I_{Kr} currents look uncharacteristic in that, there is little or no evidence for the upswing of tail current due to recovery from inactivation, prior to its exponential decay. There is no indication if this is due to the voltage you chose for displaying the current, nor do you state how you quantified I_{Kr} , which is normally measured at the tail current peak. I would like you to split Figure 4 into two (a total of 9 figures are permitted per article), with a new Figure addressing the hiPSC results and one for the hESC results, each in more detail. Please refer to Figures 1 - 2 in Sanguinetti et al. (Cell 81, p299-307) for the degree of characterization needed. It will reassure the reader that these cells generate an I_{Kr} that agrees with the literature, or if that is not the case, then please account why the current is non-classical. I would like to see I_{Kr} at two or three different hyperpolarization voltages to show the biphasic tail current, and for you to indicate the saturation of the so-called inactivating current as the depolarization voltage (first step of the two-step voltage protocol) is made more positive. It is informative to have this for both isogenic lines, hence my asking for two figures, particularly since Figure 5 shows no significant differences in any of the other measured parameters.

We thank the referee for the instructive suggestions and remarks, which helped us to present the I_{Kr} data in our hPSC-CMs much better.

The reviewer is right that there was little evidence for the upswing of tail current

due to recovery from inactivation in our typical examples. However, in our recordings, recovery from inactivation does occur, because tail current amplitudes are larger than the step currents (at positive potentials), as shown in our new Figure 4C and D and new Figure 5C and D. In the new Figures 4D and 5D current amplitudes were measured at the peak of the tail, normalised to the cell capacitance, and plotted against the test potential. The reason why recovery from inactivation was not visible in our typical examples is likely related to the fast kinetics of the recovery from inactivation, while we used voltage clamp conditions with very long voltage clamp protocols and with rather low digitizing of our signals.

According to the reviewer's suggestion, we have now split Figure 4 into two separate figures, i.e., a new Figure 4 (for hiPSC I_{Kr} results) and a new Figure 5 (for hESC I_{Kr} results). In these figures, I_{Kr} is now shown at three different depolarisation voltages (-30, -20, and -10 mV); in this way characteristic I_{Kr} currents are now more clearly visible. The depolarizing step to -10 mV and its corresponding tail current especially demonstrate that recovery from inactivation is present. To show that the tail current has a bi-exponential decay phase in more detail, we included insets in our new Figures 4B and 5B. Moreover, we now show the fast and slow deactivation time constants at four different voltages (-10, 0, +10, and +20 mV) in the revised Figure 6E and F.

It should be noted that we measured I_{Kr} in human pluripotent stem cell derived cardiomyocytes under physiological conditions, at 37°C, as E-4031 sensitive current, while in Sanguinetti et al., 1995, the HERG channel was overexpressed in *Xenopus* oocytes and the current measured at room temperature (21°C-23°C). These differences can easily account for discrepancies. However, our data are in agreement with I_{Kr} traces previously reported in human pluripotent stem cell derived cardiomyocytes (Itzhaki et al, 2011; Moretti et al., 2010; Ma et al., 2011; Lahti et al., 2012), except that we found a higher current density in hESC-derived cardiomyocytes (3.5 pA/pF) than those reported in the literature for both hiPSC-derived cardiomyocytes (0.55-1.9 pA/pF) and native human cardiomyocytes (0.25-0.6 pA/pF) (Hoekstra et al., 2012). We have now commented on this discrepancy in the "Discussion" section of the revised manuscript (page 15, last paragraph, and page 16, first paragraph).

In Figure 6 and Supplementary Figure 7, all the action potentials and field potentials are unexpectedly short in duration compared with other published accounts (Itzhaki et al. Nature 471, p225-9; Doss et al. PLoS ONE 7, p1-17). Please account for this discrepancy. Moreover, early afterdepolarizations (EADs) were not present in the prolonged action potentials (at least this is not discussed). If EADs were absent and AP prolongation was modest, please address if this is related to the observed modest loss of I_{Kr} density (30-40%). Many class 2 hERG mutations show a much greater loss of function (Anderson et al. Circulation 113, p 365-373), please account for this additional discrepancy with published findings. Is it certainly possible that the c.A2987T KCNH2 mutation is less severe than the previously reported mutations, and this is a good rationale for using isogenic lines that can detect the result of more subtle mutations. Finally, the action potentials were recorded using the perforated patch method. Please reassure the reader that this was monitored by stating all of the measures you took to be certain that patches remained perforated, and did not rupture.

We agree with the referee that both action potentials (AP) and field potentials are short in duration compared with other published data. However, the range of AP durations reported in the literature for hPSC-derived cardiomyocytes is wide, with APD₉₀ ranging from a minimum of 173 ms to a maximum of 495 ms (for a review please refer to Hoekstra et al., 2012). The AP duration that we reported in the present manuscript for the LQT2-hiPSC^{corr} cardiomyocytes falls into this range (APD₉₀=208±29 ms), while for the *NKX2.5^{eGFP/w}* hESC cardiomyocytes it is indeed shorter (APD₉₀=138±11 ms). However, it should be noted that the direct comparison between the APs in the literature is complicated by experimental differences, including variability in the cell lines used, differentiation and dissociation methods, and patch-clamp techniques. In particular, most of the studies recorded APs in single cells, while others (Doss et al., 2012; He et al., 2003) in beating clusters. Also, within single cells, most of the studies measured spontaneously beating cardiomyocytes, while one (Davis et al, 2012) recorded quiescent CMs. Moreover, the majority of the studies recorded APs between 35°C and 37°C, while others (Itzhaki et al., 2011; Zhang et al., 2011) at lower temperature (32°C and room temperature, respectively). The low temperature could be an explanation for the very long APD₉₀ reported by Itzhaki et al. Finally, most studies used the ruptured whole-cell patch-clamp technique, while a minority (included the present study) used the perforated patch-clamp technique. Taken together, all these variables may account for the variability in AP duration described in the literature. Of note, not only the hiPSC-CM measurements reported in our study are comparable with those of native human ventricular cardiomyocytes (APD₉₀=213 ms, Magyar et al., 2000), but they also are in agreement with data derived previously in our own lab (Davis et al., 2012) using the same differentiation method and the same patch-clamp technique. However, to account for the short AP measured in our hESC-CMs, we have discussed this point in the “Discussion” section, page 16, first paragraph.

Early-after depolarisations (EADs) did not appear in any of the LQT2-hPSC cardiomyocytes (LQT2-hiPSCs^{N996I} and *NKX2.5^{eGFP/w}* hESCs^{N996I}) analysed in the present study. Moreover the AP prolongation in the mutated cardiomyocytes was modest when compared with their wild-type counterparts (LQT2-hiPSCs^{N996I} vs. LQT2-hiPSCs^{corr}: APD₅₀= 227 vs. 164 ms and APD₉₀= 292 vs. 208 ms; *NKX2.5^{eGFP/w}* hESCs^{N996I} vs. *NKX2.5^{eGFP/w}* hESCs: APD₅₀= 130 vs. 104 ms and APD₉₀= 165 vs. 138 ms), especially when compared with other LQT2 hiPSC models (Itzhaki et al., 2011; Matsa et al., 2011; Lahti et al, 2012). It should be noted that in all these studies the control group consisted of wild-type hiPS cells derived from a healthy individual unrelated to the LQT2 patient from which hiPSCs were obtained, therefore over-estimation of differences cannot be excluded under these circumstances. The advantage of our approach using isogenic lines is indeed highlighted by our results, where even subtle differences could be appreciated.

In the literature it has been reported that the dominant mechanism by which missense mutations cause loss of function is to generate trafficking deficient (class 2) hERG channels (Anderson et al., 2006). We have shown that this is the disease mechanism for the missense mutation N996I that we have studied. However the location of the amino acid substitution also appears to correlate with phenotype severity, where mutations in the pore-forming region have been linked to more severe phenotypes (Moss et al., 2002). The N996I mutation is located in the C-terminal portion of the channel.

In summary, as the referee pointed out, it is clear from our study that the

c.A2987T KCNH2 mutation is less severe than other previously reported mutations, and this justified the rationale for using isogenic lines that can detect more subtle effects. We have now commented all these issues (absence of EADs, modest APD prolongation of the N996I mutated hPSC-derived cardiomyocytes compared to previously reported data, and advantage of using isogenic lines) in the Discussion, page 16, second paragraph.

In the present manuscript we used the perforated patch-clamp technique because with this method it is possible to perform experiments under conditions that show more similarity with physiological conditions (Hoekstra et al., 2012). In our experiments, we followed the access resistance continuously, and, due to incorporation of amphotericin into the membrane, the resistance lowers gradually. A rapid drop, resulting in an access resistance of $<8 \text{ M}\Omega$, is a clear indication of (unwanted) whole cell configuration and these cells die very quickly.

(2) Minor comments

Introduction, paragraph 2, the word "afflicted" is anthropomorphic and should be replaced (affected?)

We have followed the referee's suggestion and replaced the word "afflicted" with "affected" in the "Introduction" section, page 4, second paragraph.

Results, section 3, paragraph 3 (page 9), where it states that 1 μM E-4031 was used, the font looks odd. Is the number inadvertently subscripted here?

In the conversion of the word file into pdf the font was inadvertently changed because no space was present between "1" and " μM ". We have inserted a space and made sure that after the pdf file generation the font looks correct.

(3) Additional suggestions

In general, I found that the Introduction was too long and Discussion too short. I think that this manuscript might benefit from reversal of the relative lengths of these two sections. I think you might want to make the Introduction more terse, cutting out some superfluous details.

Following the referee's advice, we have now made the "Discussion" section more comprehensive by an extensive editorial revision addressing several points raised by the referees. We believe that now the balance between Introduction and Discussion is more appropriate.

Referee #2:

The manuscript by Bellin et al. described providing human cellular model of long QT syndrome type 2 (LQTS2) to investigate molecular and cellular mechanisms underlying this disease. In this study, the authors generated induced pluripotent stem (iPS) cells from patient and corrected the mutation A2987T in KCNH2 gene encoding HERG channel to generate isogenic iPS cells as control, and then differentiated the iPS cells

into cardiac cells in order to record IKr current and action potentials using electrophysiological tools, to examine gene expression using RT-PCR and to quantify glycosylated HERG proteins using Western blot. They found the mutation A2987T (N996I) in KCNH2 is the primary cause to prolong action potential in human cardiac cells in vitro. Also, they demonstrate reduction of membrane-localizing HERG channels in the patient cardiomyocytes.

Overall, I feel the study timely to understand molecular basis of LQTS using a new cellular model. Their approach could also provide a useful platform to find new ways to investigate human-specific mechanisms of cardiac arrhythmias using patient-specific pluripotent stem cells, and it would be widely informative in the fields of cardiovascular study and interesting for audience in this journal. I believe this study is basically accomplished, but it has been known that some missense mutations in KCNH2 could reduce IKr current, resulting in prolonged action potential in iPS cell-derived cardiomyocytes. In this study, unfortunately, there are no new insights into the disease although the authors introduced the same point mutation in human embryonic stem cells, generating two genetically distinct isogenic pairs of LQTS and control lines. First of all, to confirm their findings, it would be great if the authors could address two main concerns below.

We thank the referee for the largely positive comments, constructive criticism, and insights that helped us to improve the manuscript.

Major concerns:

1. In Figure 7C-D, the authors suggest that highly glycosylated HERG proteins (at 155 kDa) were reduced in cardiomyocytes expressing N996I mutant. However, the blot image of HERG protein doesn't seem to be much reliable. I believe there are some technical issues existing in the blotting steps (especially, transferring proteins in gel into PVDF membranes or incubation of the membrane to the antibodies) because 135 kDa HERG bands were not consistent among the samples (left, higher but right, lower). It'd be strongly recommended for the authors to repeat the experiments to show representative results as well as entire film image (not cutting and pasting only small area to demonstrate the bands), following recent regulation in Nature Publishing group (some journals has already asked the authors to provide whole film images as supplementary figures for all Western blots).

We thank the referee for the very useful remarks and suggestions, which helped us to improve assessment of the reduction of the complex-glycosylated 155-kDa band in the N996I-HERG channel.

At the referee's request, we repeated the Western blot for HERG on hESC-derived cardiomyocytes and replaced the blot image in the new Figure 8C (old Figure 7C). The blot shown in the figure is now representative of n=4 independent biological repeats. Densitometric quantification of the 155 kDa as well of the 135 kDa bands is based on these four replicates, as stated in the Figure Legend "values are presented as mean \pm s.e.m., n=4". We have now provided in the Supplementary Information all four blots including their whole film image. These results clearly show a consistent and

reproducible reduction of the complex-glycosylated band in the cardiomyocytes harbouring the N996I-HERG mutation. We have also calculated HERG trafficking efficiency as ratio of the fully glycosylated 155 kDa band over total HERG protein (155 kDa band + 135 kDa band). Compared to the wild-type cells, a reduction of ~40% was estimated in the mutated cardiomyocytes, as now specified in the Results, page 11, last paragraph, and shown in the new Supplementary Figure S10A.

Furthermore, we have performed two more Western blot experiments on hiPSC-derived cardiomyocytes. These new data have also been included in the new Figure 8 where we show a representative image (Figure 8E) and the average densitometric quantification from the two blots (Figure 8F) as stated in the Figure Legend “values are presented as mean \pm s.e.m., n=2”. In the Supplementary Information we have provided the whole film images. Of note, the data gathered from the hiPSC-CMs suggested that in the mutated cells the 155 kDa band, corresponding to the complex-glycosylated HERG channel, is also reduced compared with the corrected CMs, resulting in a decrease of HERG trafficking efficiency of ~30%, as now shown in the new Supplementary Figure S10B. The difference in the values of HERG trafficking efficiency between the mutated, FACS sorted hESC-CMs and the patient-specific, manually-microdissected hiPSC-CMs might result from a higher percentage of non-cardiomyocyte cells in the latter group.

It is important to point out that, to our knowledge, this is the first time that HERG channel protein could be detected by Western blotting in human cardiomyocytes derived from hPSC under physiologically relevant conditions. Previous published papers using hiPSCs to model LQT2S (Itzhaki et al., 2011; Matsa et al., 2011; Lahti et al., 2012) did not show HERG protein by Western blot unless the channel was overexpressed in fibroblasts and not in cardiomyocytes (Matsa et al., 2013).

2. Although the authors mention that "Furthermore, the reduction of the mature complex-glycosylated 155 kDa protein band in the mutated hESC-CMs suggests that the mutation results in a misfolded protein that does not reach the Golgi and, most likely, the cell membrane", in this study the authors don't show any evidences to support this molecular phenotype. It would be required to examine if the proteins are aggregated in the ER/SR or the Golgi and if it could induce ER stress. Also, the mutation might affect some associating molecules that are essential for HERG trafficking/internalization, ubiquitination or protein stability. Providing novel insights into LQTS2 would be necessary rather than showing expected results such as prolonged action potential and reduced IKr current.

We thank the referee for insightful comments and constructive remarks that encouraged us to analyse the membrane-fraction channel reduction phenotype of the N996I-mutated HERG protein in our human pluripotent stem cell model further. Following the referee's suggestion, we investigated whether the mutated proteins are aggregated in the ER/SR or the Golgi by performing single cell immunofluorescence analysis of HERG in combination with the ER marker PDI or the Golgi marker golgin-97 (Supplementary Figures S8 and S9). We did not observe any clear difference in the subcellular distribution of HERG channels between CMs generated from the LQT2-hPSCs^{N996I} models and their wild-type or corrected counterparts, because some HERG channel signal was detected in the ER and in the Golgi (compartments where the channel normally

transits) also in the latter group. Furthermore, Western blot experiments did not show an average increase of the 135 kDa band (corresponding to the core-glycosylated protein that is present in the ER, see new Figure 8D and F), suggesting that most likely the N996I mutation does not induce aggregation/accumulation of HERG protein in the ER. We further proceeded to investigate whether the ER stress pathway was induced in the mutated CMs by looking at Activating Transcription Factor 6 (ATF6), a key regulator of transcriptional control in the unfolded protein response (UPR) (Chen et al., 2002; Haze et al., 1999). In particular, UPR triggers the cleavage of ATF6 into its activated form, which in turns upregulates the synthesis of ER chaperones, such as Calnexin and Calreticulin. This mechanism of induced ER stress and clearance pathways has been already associated with some HERG mutations (Guo et al., 2012; Smith et al., 2011; Wang et al., 2012). Western blot for ATF6 revealed the presence of several forms of ATF6, including both the unprocessed protein embedded in the ER (90 kDa band) and some smaller forms resulting from the cleavage of the full-length ATF6 during ER stress, with no detectable differences between wild-type and mutated *NKX2.5-eGFP⁺* cells (Figure 9A). Furthermore, Calnexin and Calreticulin were also similarly expressed in both wild-type and mutated hESC-CMs (new Figure 9B and C). We concluded that the N996I-HERG mutation does not elicit the UPR.

To assess the involvement of proteasome-mediated and/or lysosome-mediated degradation of HERG channel, we analysed the effects of the proteasome inhibitor lactacystin and of the lysosome inhibitor leupeptin on the steady state of HERG protein levels in wild-type and mutated hESC-CMs. As now described in the Results section, page 12, we purified the *NKX2.5-eGFP⁺* cell population by flow cytometry, treated the cells for 24 hours with either lactacystin (20 μ M) or leupeptin (100 μ M) and then performed immunoblotting of the cell lysates with anti-HERG antibody (new Figure 9D). Treatment with the proteasome inhibitor lactacystin augmented the level of the mature, highly glycosylated form of HERG in *NKX2.5^{eGFP/w} hESC^{N996I} eGFP⁺* cells, resulting in a \sim 1.4 fold increase of channel trafficking efficiency. No increase was observed in the wild-type counterparts or after treatment with the lysosome inhibitor leupeptin (Figure 9D and E), suggesting a role of proteasomes in the degradation of N996I HERG protein. Further analysis is required to unravel the exact/additional mechanism through which mutated HERG channels are processed (retained in the ER/degraded through proteasomes/degraded through lysosomes), but this goes beyond the scope of this paper. It is important to point out that most of our knowledge about the mechanism of altered behaviour of mutated HERG channels comes from overexpression studies in heterologous systems. When compared to a heterologous system, the use of hPSC-derived CMs is associated with many technical challenges related to the process of cell differentiation and CM isolation. Thus, a maximum of 60-80 plates of differentiating hPSCs (\sim 30-40 plates/cell line, the wild-type and the mutated cells) could be processed per day, dissociated and finally FACS sorted. Each experiment allowed the isolation of $1-5 \times 10^6$ eGFP+ cells, with a total amount of purified protein ranging from 60 to 250 μ g. Therefore, during the revision time (3 months), we could set up only 8 independent experiments for Western blotting analysis. Nevertheless, we believe that the results obtained on the mechanistic of N996I-HERG channel trafficking in hPSC-derived CMs are indeed very informative, considering that in every overexpression heterologous system non-physiological levels of both wild-type and mutated proteins could alter

molecular stoichiometry. Since this is the first time that such experiments have been carried out in a human physiologically relevant system, we have presented our data in a new Figure 9.

In summary, we agree with the referee that human pluripotent stem cell based systems will be helpful in the future to investigate and confirm data obtained in overexpression systems; however this was not the main goal of our study. Our aim was to investigate, under genetically defined conditions, the electrophysiological characteristics of the N996I mutation that seems to have a mild phenotype and therefore would undoubtedly benefit in accuracy from the use of isogenic pairs. Therefore we have modified the “Discussion” section, page 18, where we speculated that “Furthermore, the reduction of the mature complex-glycosylated 155 kDa protein band in the mutated hESC-CMs suggests that the mutation results in a misfolded protein that does not reach the Golgi and, most likely, the cell membrane” with “Furthermore, the reduction of the mature complex-glycosylated 155 kDa protein band in the mutated hPSC-CMs suggests that, due to the mutation, a smaller number of channels are present at the cell membrane. This could be partially due to proteasome targeting for degradation, but might not be the only mechanism for this N996I specific mutation. Further studies are required to assess more in detail the exact contribution of proteasome clearance pathway, protein channel stability, and lysosome degradation.” In addition, we have also modified the sentence at page 14 “Furthermore, the reduction of the fully glycosylated mature 155 kDa protein band detected by Western blot in the mutated hESC-CMs suggested that the mutation might cause protein misfolding and result in defective channel trafficking (Vandenberg et al, 2012)” that now reads “Furthermore, the reduction of the fully glycosylated mature 155 kDa HERG protein in the mutated hPSC-CMs and its partial rescue after treatment with the proteasome inhibitor lactacystin suggest that the mutation might induce, to some extent, protein clearance through the proteasomes, thus resulting in defective channel trafficking (Vandenberg et al, 2012). Further analysis of HERG protein stability and internalization is required to unravel the additional mechanisms through which mutated N996I HERG channels may be processed.”

Minor points:

1. Combining Fig. 1 and 2 would be appropriate.

We thank the referee for this appropriate suggestion. However, because a total of 9 figures per article are permitted, we kept Figures 1 and 2 separated.

2. Teratoma formation assays would be required since in this study the gene expression profiling of in vitro spontaneous differentiation was very noisy. For instance, LQT2-hiPSCs-N996I showed 10-fold higher expression of PTF1A, CD31, MYL2 and TH genes than LQT2-hiPSC-corr although both Nkx2.5-GFP lines showed similar expressions.

We thank the referee for this comment. We recognise that teratoma formation is widely regarded as a stringent assay for pluripotency in human pluripotent stem cells; however PluriTest, a microarray based analysis benching marking cells to a panel of standardized pluripotent stem cells, has been recently described as a robust and open access

bioinformatic assay of pluripotency in human cells based on their gene expression profiles (Müller et al., 2011). We collected RNA from undifferentiated LQT2-hiPSCs^{N996I}, LQT2-hiPSCs^{cor}, and *NKX2.5^{eGFP/w}* hESCs^{N996I}, performed microarray analysis and ran the PluriTest (<http://www.pluritest.org>). All the three lines had a high pluripotency score (indicating that they contain a pluripotency signature) and a low “novelty score”, indicating that they are indistinguishable from normal human pluripotent stem cells. These results are now presented in the new Supplementary Figure S6 of the revised manuscript and described in the “Results” session, page 8, and in the “Material and methods” session, page 19.

3. In Figure 6A,C, there are some artifacts in depolarization phase of the action potentials.

In our experiments, we used slow spontaneously contracting cells that were paced at the stimulation frequency of 1 Hz. In the new Figure 7A and C (old Figure 6A and C), the depolarization phase shows the injection of current used to stimulate the cells at the indicated frequency.

Referee #3:

*Bellin and colleagues present a manuscript examining a mutation in KCNH2 linked to the long QT syndrome using isogenic human pluripotent stem cells. Specifically, the authors study the A2987T (N996I) mutation in KCNH2 in isogenic induced pluripotent (iPS) cell lines and isogenic embryonic stem (ES) cell lines. The authors generated a patient-derived iPS cell line harboring the N966I mutation and performed genetic correction by homologous recombination to produce an isogenic line for comparison. In complementary studies, the authors introduced the N966I mutation into a wild type hESC reporter line ESC *Nkx2.5eGFP*. In both cases, the N966I mutation decreased *I_{Kr}* and prolonged action potential duration relative to the isogenic control. Furthermore, the authors provide evidence that the decreased current is due to a defect in trafficking of the mutant protein to the surface membrane. The strength of the study is that this is the first example to my knowledge using isogenic iPS and ES cell lines to demonstrate clearly and consistently a disease causing mutation in an inherited heart disease. There are some curiosities in the data such as the differences in the action potential properties comparing the iPS cell- and ES cell-derived cardiomyocytes. It seems that the ES cell-derived cardiomyocytes exhibit a more immature electrophysiological phenotype. Nevertheless, the N966I mutation prolongs the APD and decreases *I_{Kr}* in both cases. Overall, by overcoming the limitations associated with modifying genes and family controls, the authors provide clear results with the isogenic cell lines. The study is presented well, and the data are compelling to this reviewer. I have only minor technical questions as follow:*

1) The methods state that electrophysiological studies were performed on cells 20-60 days in culture. This is quite a broad range, and other studies suggest that the cardiomyocytes exhibit a changing phenotype with some maturation over this time period. Does this range of differentiation time impact potentially on the differences

observed between the iPSC cells and ES cells?

We thank the referee for appreciating our work and approach to studying this KCNH2 mutation using isogenic pairs of human pluripotent stem cells.

We agree that 20-60 days culture is a wide time range for collecting cardiomyocytes. Indeed it is known that time in culture, and specifically time after the onset of spontaneous contraction, is a major factor affecting cardiomyocyte structure, intracellular calcium stores, and ion channel expression that impact the action potential (Robertson et al., 2013). However, different elements of maturity (structure, proliferation, metabolism, and electrophysiology) seem to be affected by line (Zhang et al., 2009), co-cultured cells (Kim et al., 2010), culture conditions (Mummery et al. 2012), and not only by time in culture (Zhang et al., 2009). In a simple-minded but pragmatic way, hPSC-derived cardiomyocytes have been classified as “early” CMs (<21 days of differentiation), and “late” CMs (>35 days of differentiation) (Robertson et al., 2013). In our study, by using cells in the time window of 20-60 days, we have excluded the “early” phase hPSC-CMs and performed our experiments in the time range in which most of the studies agree on the degree of maturation. Most importantly, the same time frame of 20-60 days was used for both hiPSC and hESC-CMs. Therefore, the differences that we have seen between hiPSC and hESC CMs must be more inherent to the cell lines (e.g. their origin) than to the time window.

Certainly the mechanism by which maturity changes during *in vitro* culture is incompletely understood, and other important factors remain largely unknown, since even after several months of culture, cardiomyocytes remain embryonic in phenotype and never reach adult characteristics (Zhang et al., 2009).

2) In order to allow comparative studies in the future, it would be useful to know the average whole cell capacitances for the cells patched.

We agree with the referee that these data could be informative, therefore we have now indicated the average whole cell capacitance in the Supplementary Information of the revised manuscript, namely in Supplementary Figure S11. The average whole cell capacitance was 40.8 ± 4.2 and 32.9 ± 3.1 pF in the hiPSC-CM group (LQT2-hiPSCs^{N996I} and LQT2-hiPSCs^{corr}, respectively) and 21.0 ± 3.1 and 18.4 ± 1.8 pF in the hESC-CM group (*NKX2.5*^{eGFP/w} hESCs and *NKX2.5*^{eGFP/w} hESCs^{N996I}, respectively). While the average whole cell capacitance was different between the hiPSC and the hESC groups (being larger in the hiPSC-CMs than the hESC-CMs) no significant difference was detected within the two separate groups.

3) p. 7, line 5, "... resulting in an isoleucine to asparagine substitution at position 996 (N996I)..." should be "... resulting in an asparagine to isoleucine substitution at position 996 (N996I)..."

We have corrected this sentence that now reads “resulting in an asparagine to isoleucine substitution at position 996 (N996I)...”.

Thank you for submitting your revised dataset that has been re-evaluated by one of the original referees. While appreciating the amount of additional work that went into the revised data and thus essentially supportive of eventual publication here, some concerns remain on the statistical tests applied and therefore the reliability of some of the presented data.

I kindly ask you to carefully consider the very constructive remarks aimed at resolving these issues and provide us with a finally amended version of your study to your earliest convenience.

Please notice that according to our policy of 'scooping protection' potential competing work published while you are revising your data will NOT impinge on the proceedings of your work, at least within a reasonable timeframe. Please invest the necessary time, in case further experimental work is needed, and provide us with a specific point-by point response to facilitate rapid proceedings of the ultimate version of your manuscript.

Thank you for the opportunity to consider your work for publication. I look forward to your final amendments.

REFEREE REPORTS:

The revised version of the manuscript is substantially improved and most of the suggestions have been well met.

Concern remains about the action potential data presented in Figure 7.

Major Concerns.

(1) Inclusion of the details of p-values and statistical tests as required by the EMBO journal reveals that the t-test was used to compare APDs in Figure 7B, while the non-parametric U-test (Mann-Whitney) was used for the identical purpose in Figure 7D. This implies a weakness in the data which should be resolved. To my knowledge, the most appropriate test to use is the independent t-test making an assumption of unequal variances, which should be applied to both data sets. This assumes normality in the data, and if this is an unreliable assumption, a test of normality is required. If the results are non-normally distributed the U-test ought to be applied to both data sets (there seems to be no a-priori reason why iPSC APDs should be normally distributed and ESC APDs not, for an identical metric). It is an additional concern that p-values assigned to these histograms are borderline, especially in Figure 7D. The implication is that the mean hERG deficit determined by voltage clamp was insufficient to affect APD which calls the validity of the entire model into question. Strictly, to resolve this, hERG expression and APD ought to be determined in the same myocyte, or a pharmacological investigation, along the lines of titration of % hERG block vs APD,

needs to be carried out.

(2) Certain literature, (e.g. Doss et al. PLOS ONE 7(7) 2012) suggests that IKr density strongly influences the resting potential in stem cell-derived cardiomyocytes owing to a lack of expression in IK1 channels. One might expect that the 30% - 40% fall in peak hERG current densities in cells affected by the N996I mutation should have also depolarized the MDP (since you hypothesize this deficit underlies the differing APDs), yet this clearly was not observed. Moreover, the hERG current densities in N996I ESC-cardiomyocytes and in the corrected iPSC-myocytes were practically identical (2 pA/pF), yet the measured MDPs differed considerably between these cells. This inconsistency needs to be addressed.

Based on these two points, it is an awkward concern that the membrane potential data from the hESC-derived myocytes was in some way confounded. Also, given that the N value was quite large (28), the authors are encouraged to revisit these results for further scrutiny to try to account for the aforementioned inconsistencies in order to satisfy this reviewer.

Minor Points

ABSTRACT

Paragraph 2, sentence 1 - you might want to mention that the KCNH2 mutation is associated with LQT2

Paragraph 3, sentence 1 - the word "regulate" in relation to ion channel current seems inappropriate

INTRO

Paragraph 1, sentence 2 - I suggest replacing the word "changes" for "phenotypes"

Paragraph , final sentence - "pathomechanism" is jargon, I suggest using "pathological mechanism"

RESULTS

Paragraph 5, sentence 1 & sentence 6 (page 9) - It looks like the terminology is inconsistent: are "LQT2-hiPSCs-N996I/loxP-control" and "loxP-control LQT2-hiPSC-N996I" the same thing? I suggest using one simplified name throughout the text.

POINT-BY-POINT RESPONSE TO THE REFEREE

We thank the referee for appreciating the substantial improvements made during the revision and for the constructive criticisms and comments. We have addressed each of the two major issues by carefully checking the statistics in all datasets throughout our paper, by asking an expert statistician to evaluate the statistics used and highlighting specific details in our results. Editorial revision also addressed major and minor concerns. We believe these clarify key issues highlighted in the review. These changes have been incorporated in the revised version of the manuscript. The new manuscript text and figures, as well as a point-by-point rebuttal are provided for the referees and editor.

Referee #1:

The revised version of the manuscript is substantially improved and most of the suggestions have been well met.

Concern remains about the action potential data presented in Figure 7.

Major Concerns.

(1) Inclusion of the details of p-values and statistical tests as required by the EMBO journal reveals that the t-test was used to compare APDs in Figure 7B, while the non-parametric U-test (Mann-Whitney) was used for the identical purpose in Figure 7D. This implies a weakness in the data which should be resolved. To my knowledge, the most appropriate test to use is the independent t-test making an assumption of unequal variances, which should be applied to both data sets. This assumes normality in the data, and if this is an unreliable assumption, a test of normality is required. If the results are non-normally distributed the U-test ought to be applied to both data sets (there seems to be no a-priori reason why iPSC APDs should be normally distributed and ESC APDs not, for an identical metric). It is an additional concern that p-values assigned to these histograms are borderline, especially in Figure 7D. The implication is that the mean hERG deficit determined by voltage clamp was insufficient to affect APD which calls the validity of the entire model into question. Strictly, to resolve this, hERG expression and APD ought to be determined in the same myocyte, or a pharmacological investigation, along the lines of titration of % hERG block vs APD, needs to be carried out.

We have carefully checked all of the statistics in all datasets throughout the paper, with special attention paid to the APD measurements presented in Figure 7.

We performed all our statistical analysis using the SigmaStat 3.5 software (<http://www.sigmaplot.com/products/sigmaplot/sigmastat.php>). This program allows testing of the normality and equality of the datasets prior to application of a statistical test. When both conditions were fulfilled, unpaired Student's t-test was applied; when one of these two conditions was not fulfilled, the non-parametric Mann-Whitney test was applied. $P < 0.05$ was considered statistically significant. These details are now included in the "Material and methods" section, "Statistical analysis" paragraph, page 24.

In the legend of Figure 7 we unfortunately made an error in the text; in fact the unpaired Student's t-test was applied to datasets of both Figure 7B and 7D. We

apologize for missing this mistake and we thank the referee for pointing out the incongruence. The hiPSC-CM APD₅₀ and APD₉₀, and hESC-CM APD₅₀ data were normally distributed and had equal variances; therefore the unpaired Student's t-test was applied. Statistically significant differences (P<0.05) were observed and the corresponding p-values, together with the applied t-test, are reported in Figure Legend 7B and D. hESC-CM APD₉₀ measurements were not normally distributed; however, relying on the central limit theorem, means are normally distributed for "large N", irrespective of the underlying distribution, and we applied the t-test anyway, thereby using the same test for all p-values in figure 7B and 7D (large N is roughly N>30, see e.g. Medical Statistics at a Glance, Petrie & Sabin, 2006). In the case of hESC-CM APD₉₀, no statistically significant difference was observed. As we described in the "Results" section, we concluded that in hiPSC-CMs I_{Kr} reduction caused a significant prolongation of both APD₅₀ and APD₉₀, while in the hESC model, only the APD₅₀ of the mutated CMs was significantly prolonged compared with the wild-type CMs.

A pharmacological investigation of hERG block titration (using increasing concentrations of E-4031 in the range of 300 pM-30 μM), along with field potential duration (FPD) was already presented in the original version of our manuscript in Supplementary Figure S7 for both hiPSC- and hESC-CMs. As described in the "Results" section, this figure shows a dose-dependent FPD prolongation in CMs derived from all the hPSC lines used in the present work, indicating the presence of HERG channel. Specifically, the FPD prolongation was larger in the mutated CMs when compared to their wild-type counterparts (both hiPSC- and hESC-CMs), indicating that the N996I mutation conferred increased sensitivity to I_{Kr} blockade (Figure S7A and C). In line with the milder effect of I_{Kr} reduction on APD prolongation in hESC-CMs compared with hiPSC-CMs as assessed by single-cell electrophysiology, MEA measurements also showed that the FPD prolongation was less pronounced in the hESC-CMs than in hiPSC-CMs. However, the FPD change between mutated and wild-type CMs was similar (~60-70%) at E-4031 concentrations ≥300 nM for both hiPSC and hESC myocytes (Figure S7B and D, showing quantification at 30μM E-4031). These results suggest an exclusive effect of the N996I mutation on the electrophysiological phenotype in both hiPSC- and hESC-CMs and support the robustness of our model.

(2) Certain literature, (e.g. Doss et al. PLOS ONE 7(7) 2012) suggests that IKr density strongly influences the resting potential in stem cell-derived cardiomyocytes owing to a lack of expression in IK1 channels. One might expect that the 30% - 40% fall in peak hERG current densities in cells affected by the N996I mutation should have also depolarized the MDP (since you hypothesize this deficit underlies the differing APDs), yet this clearly was not observed. Moreover, the hERG current densities in N996I ESC-cardiomyocytes and in the corrected iPSC-myocytes were practically identical (2 pA/pF), yet the measured MDPs differed considerably between these cells. This inconsistency needs to be addressed.

We thank the referee for this comment and we are indeed aware of the limitations of hPSC-derived cardiomyocytes for cardiac disease modelling that are especially related to the immaturity of these cells, including the lack of a proper I_{K1}, which is always lower than that measured in native ventricular cardiomyocytes, regardless of the length of time the cells are kept in culture (Doss et al., 2012; Blazeski et al., 2012). In line with this, even if in Figure 3B we do show the expression of *KCNJ12* (one of the molecular contributors to I_{K1}) both in CMs derived from hiPSCs and in the

eGFP+ cell population derived from hESCs, we do not know at which level the specific current is indeed present. Most probably, much like most of the published hPSC-derived CMs, I_{Kr} is relatively small in our cells compared to human adult ventricular myocytes. Under these circumstances, it is crucial to highlight that we have demonstrated that the N996I mutation causes a 30-40% reduction of I_{Kr} . In contrast, Doss et al. (2012) used E-4031 at a concentration of 5 μ M, allowing for a complete block of I_{Kr} . This difference could be one of the reasons that we do not see a depolarized MDP in the CMs harbouring the N996I mutation. In support of our findings, other studies using LQT2-hiPSC-CMs did not report any depolarization in the MDP (Itzhaki et al., 2011; Matsa et al., 2011), even in presence of more severe mutations. Therefore, even if computational models (e.g. Doss et al., 2012) would suggest that I_{Kr} reduction might affect the MDP, experimental data show that this happens only sporadically. Clearly, further investigation is needed to resolve this issue, and more cell lines, with various differentiation methods and media should be analysed in order to draw more general conclusions.

As the referee pointed out, we are aware that the hERG current densities in the N996I ESC-CMs and in the corrected iPSC-CMs were similar (\sim 2 pA/pF), while the MDPs differed between these lines. Again, this observation suggests that in our system, the differences observed in the MDP are minimally, if at all, dependent on I_{Kr} . Most likely, this specific discrepancy is part of a wider dissimilarity in the AP characteristics between our hiPSC and hESC lines including different origins, differentiation properties, or epigenetic factors, as already mentioned in the “Discussion” section. For this reason we never intended to directly compare hiPSC and hESC data, but instead studied the specific effect of the mutation within the same cell line, under genetically defined conditions. It would require many more cell lines in a “population” type of approach to make statements about differences between hiPSC-CMs and hESC-CMs.

Based on these two points, it is an awkward concern that the membrane potential data from the hESC-derived myocytes was in some way confounded. Also, given that the N value was quite large (28), the authors are encouraged to revisit these results for further scrutiny to try to account for the aforementioned inconsistencies in order to satisfy this reviewer.

Minor Points

ABSTRACT

Paragraph 2, sentence1 - you might want to mention that the KCNH2 mutation is associated with LQT2

We have included this suggestion in the abstract, paragraph 2; the sentence now reads: “To study the LQT2-associated c.A2987T (N996I) *KCNH2* mutation under genetically defined conditions, we derived iPSCs from a patient carrying this mutation and corrected it.”

Paragraph 3, sentence1 - the word "regulate" in relation to ion channel current seems inappropriate

We have replaced the word “regulate” with “conducted”; the sentence now reads: “Correction of the mutation normalized the current (I_{Kr}) conducted by the HERG channel and the action potential duration in iPSC-derived cardiomyocytes.”.

INTRO

Paragraph 1, sentence2 - I suggest replacing the word "changes" for "phenotypes"

We have now replaced the word “changes” with “phenotypes”.

Paragraph, final sentence - "pathomechanism" is jargon, I suggest using "pathological mechanism"

We have now replaced the word “pathomechanism” with “pathological mechanism”.

RESULTS

Paragraph 5, sentence1 & sentence 6 (page 9) - It looks like the terminology is inconsistent: are "LQT2-hiPSCs-N996I/loxP-control " and "loxP-control LQT2-hiPSC-N996I " the same thing? I suggest using one simplified name throughout the text.

We apologise for this inconsistency and we have now used the name “LQT2-hiPSCs^{N996I/loxP-control}” throughout the manuscript.